# Molecular mechanisms of ion conduction and ion selectivity in TMEM16 lipid scramblases

Andrei Y. Kostritskii [1,2,3] & Jan-Philipp Machtens [1,2✉]

TMEM16 lipid scramblases transport lipids and also operate as ion channels with highly variable ion selectivities and various physiological functions. However, their molecular mechanisms of ion conduction and selectivity remain largely unknown. Using computational electrophysiology simulations at atomistic resolution, we identified the main ion-conductive state of TMEM16 lipid scramblases, in which an ion permeation pathway is lined by lipid headgroups that directly interact with permeating ions in a voltage polarity-dependent manner. We found that lipid headgroups modulate the ion-permeability state and regulate ion selectivity to varying degrees in different scramblase isoforms, depending on the amino-acid composition of the pores. Our work has defined the structural basis of ion conduction and selectivity in TMEM16 lipid scramblases and uncovered the mechanisms responsible for the direct effects of membrane lipids on the conduction properties of ion channels.

[1] Institute of Biological Information Processing (IBI-1), Molekular- und Zellphysiologie, and JARA-HPC, Forschungszentrum Jülich, Jülich, Germany. [2] Institute of Clinical Pharmacology, RWTH Aachen University, Aachen, Germany. [3] Department of Physics, RWTH Aachen University, Aachen, Germany. ✉email: j. machtens@fz-juelich.de

ipid scramblases of the TMEM16 (anoctamin) family are membrane proteins that facilitate the passive transport of lipids across the membrane in response to an increase in intracellular $Ca^{2+}$ concentration. The TMEM16 family also includes $Ca^{2+}$-activated $Cl^-$ channels (TMEM16A and TMEM16B)[1–3], and some TMEM16 lipid scramblases have also been shown to conduct ions[4]. Examples of such bifunctional TMEM16 proteins include mammalian TMEM16F[5–8] and TMEM16K[9,10], and the fungal homologs afTMEM16[11] and nhTMEM16[12,13]. Ion currents mediated by scramblases have been implicated in various physiological functions including apoptotic cell shrinkage[14,15], innate immunity[16], and regulation of $Ca^{2+}$ signaling[5,17]. Importantly, the physiological effects of scramblase-mediated ion currents critically depend on their ion selectivity; however, this property is highly variable between different studies[4,18]. In particular, the TMEM16F lipid scramblase has been variously reported to generate nonselective[19,20], anion selective[6,7,14], and cation selective[5,21,22] currents. In addition, a single mutation was shown to convert TMEM16A (a bona fide $Cl^-$ channel) into a cation-selective lipid scramblase[13]. Finally, the fungal scramblases (as well as TMEM16K) exhibit essentially nonselective permeability to ions when reconstituted into liposomes[10–12]. However, the structural determinants of ion selectivity in TMEM16 lipid scramblases and the precise molecular mechanisms of ion conduction remain unknown, preventing a full understanding of the physiological roles of TMEM16 lipid scramblases.

The crystal structure of the fungal homolog in *Nectria haematococca* (nhTMEM16) provided the first structural insights into TMEM16 family proteins[23] and led to the discovery of key molecular details of the scrambling process[13,24–26]. Structural analysis of nhTMEM16 revealed a dimeric butterfly-like scaffold

(Fig. 1a), which was later confirmed in other TMEM16 isoforms[10,22,27–30]. In the fully activated $Ca^{2+}$-bound state, each protomer forms a so-called subunit cavity (i.e. a membrane-spanning hydrophilic groove) that is exposed to the hydrophobic membrane environment (Fig. 1b). This unique topological feature has been assumed to underlie the lipid-scrambling mechanism by providing a pathway for polar lipid headgroups to move across the membrane. Molecular dynamics (MD) simulations showed that the lipid tails are retained within the hydrophobic interior of the bilayer upon crossing the membrane[10,13,24–26,31], consistent with the proposed "credit card" mechanism of lipid scrambling[32]. Notably, as demonstrated by recent cryo-electron microscopy (cryo-EM) studies, $Ca^{2+}$-bound nhTMEM16 in lipid nanodiscs is in conformational equilibrium between the fully open, intermediate, and closed states of the subunit cavity, with the different states presumably corresponding to different steps of scramblase activation[33].

Based on the assumption that ions permeate through the pore formed by the subunit cavity, two main models of the ion conduction have been proposed so far. First, the "alternating pore-cavity" model[22] was suggested based on an intermediate, semi-closed conformation of the cavity that was observed in several cryo-EM structures of $Ca^{2+}$-bound TMEM16 lipid scramblases[22,25,27,33]. In this conformation, the extracellular part of the cavity is sealed from the membrane, so the structure was supposed to represent an ion (but not lipid) conductive state (Fig. 1c). Second, the proteolipidic-pore model assumes that ions and lipids are transported via a single protein conformation, in which the ions permeate through a pore formed by the open subunit cavity and lined by lipid headgroups[34] (Fig. 1b). Although recent MD simulations support ion conduction through the proteolipidic pore[13], results of structural and

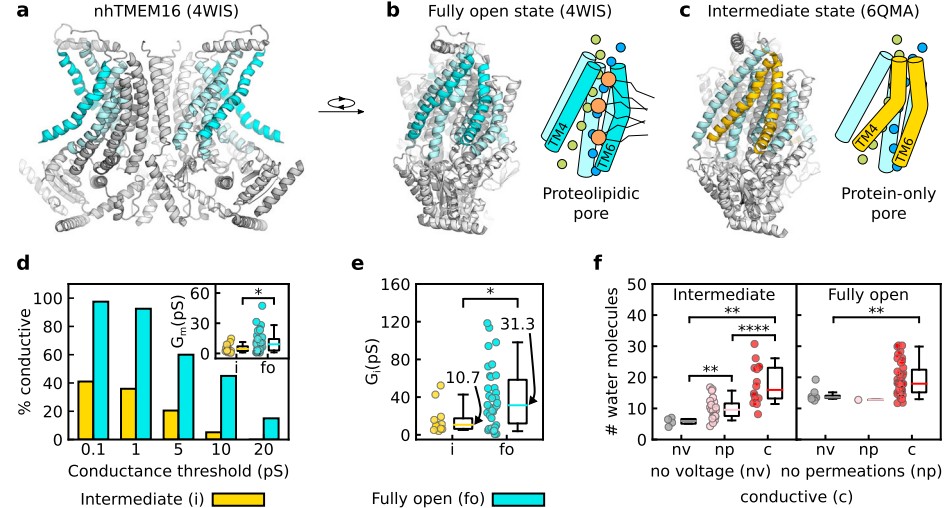

**Fig. 1 nhTMEM16 assumes the main ion-conductive state in the fully open conformation. a** Dimeric structure of the nhTMEM16 lipid scramblase, with helices lining the subunit cavity shown in cyan. **b, c** Fully open (**b**) and intermediate (**c**) conformations of the subunit cavity (represented by the 4WIS and 6QMA structures, respectively), together with schematics of the corresponding ion conduction models. Transmembrane domains (TM) 4 and 6 bordering the subunit cavity are indicated. Ions are shown as green and blue spheres and lipids are represented by orange headgroups and black tails. **d** Dependence of the percentage of conductive protomers in simulations of intermediate or fully open structures of nhTMEM16 on the mean-conductance threshold used to distinguish conductive and nonconductive states. The inset shows the mean conductance calculated globally as ratio between number of permeation events and the simulation time since a first permeation event. $n_i = 16$, $n_{fo} = 39$ independent protomers, $p = 0.03$. **e** Instantaneous conductance of nhTMEM16 in the intermediate (i) and fully open (fo) conformations. $n_i = 16$, $n_{fo} = 39$ independent protomers, $p = 0.02$. Median values are labeled. **f** Pore hydration (number of water molecules in the extracellular part of the subunit cavity) was calculated separately for conductive and nonconductive (no permeations) nhTMEM16 protomers and protomers in the absence of voltage (no voltage). Intermediate: $n_{nv} = 4$, $n_{np} = 23$, $n_c = 16$ independent protomers, $p_{nv,np} = 0.006$, $p_{nv,c} = 0.002$, $p_{np,c} = 0.00003$; Fully open: $n_{nv} = 8$, $n_{np} = 1$, $n_c = 39$ independent protomers, $p_{nv,c} = 0.002$. **d–f** Each data point represents an independent protomer, and boxplots are defined as follows: the middle line is the median, the lower and upper hinges correspond to the first and third quartiles, whiskers show the 5th and 95th percentiles. Significance was evaluated with the Mann–Whitney test, one-sided: *$p < 0.05$, **$p < 0.01$, ****$p < 0.0001$.

functional studies[25,33] propose the intermediate semi-closed conformation as the main conductive state of TMEM16 lipid scramblases. Thus, results from structural, functional, and computational studies have not yet reached a consensus on the mechanism of ion conduction in TMEM16 lipid scramblases.

To fill the gap between static structural and macroscopic functional studies, we employed extensive Computational Electrophysiology (CompEL)[35,36] MD simulations of nhTMEM16, human TMEM16K, and murine TMEM16F to gain a dynamic atomic-level picture of the ion conduction mechanisms of TMEM16 lipid scramblases. We refuted the "alternating pore-cavity" model by demonstrating that both intermediate and fully open states of nhTMEM16 are ion conductive. Importantly, our results established the fully open pore as the main ion-conductive state. We demonstrated that TMEM16 scramblases in the fully open state conduct ions through a structured, but dynamic, proteolipidic pore. We then elucidated how lipids lining the nhTMEM16 pore influence its ion permeability and selectivity. Finally, we showed that TMEM16K conducts cation-selective currents and described the molecular determinants of its ion selectivity. Taken together, our results provide a novel perspective on the experimentally observed functional properties and physiological functions of TMEM16 lipid scramblases.

## Results

**The fully open conformation represents the main ion-conductive state of nhTMEM16.** To study ion conduction of TMEM16 lipid scramblases we conducted MD simulations on different states and iosoforms of fungal and mammalian scramblases using the CHARMM force field[37,38]. In our simulations we observed hundreds of ion permeation events (see Source Data file), which were induced by electrochemical gradients and consequent transmembrane voltages generated using the CompEL method[35,36]. Briefly, a simulation system contained two membranes that separated two compartments in a periodic simulation box (Supplementary Fig. 1). The transmembrane voltage was set by a charge imbalance, which was sustained by the CompEL algorithm that swaps ions and water molecules once ion permeation has taken place[35]. Thus, the method is conceptually similar to electrophysiological experiments, where the transmembrane voltage is controlled by releasing/adsorbing $Cl^-$ ions to/from the solution by Ag/AgCl electrodes. Each membrane contained an nhTMEM16 dimer, which was subjected to either positive or negative transmembrane voltage. Unless otherwise stated, a 1-palmitoyl-2-oleoyl-phosphatidylcholine (POPC) membrane and 250 mM NaCl bulk salt concentration was used.

To identify the main ion-conductive state of scramblases, we simulated dimeric nhTMEM16 in the fully open (PDB ID: 4WIS) and intermediate (PDB ID: 6QMA) conformations. Individual simulations were run for 0.5–1 μs and the applied voltages ranged from 300 to 750 mV (Supplementary Tables 1 and 2). Since TMEM16 protomers function independently[39,40], multiple simulation replicas enabled us to collect statistics from up to 22 independent protomer trajectories (see Supplementary Tables 1 and 2). $Na^+$ and $Cl^-$ permeations induced by the applied voltage were observed in both the intermediate and the fully open states, with ions permeating exclusively along the subunit cavities, in accordance with the previously reported MD simulations of nhTMEM16 in the fully open state[13]. However, the probability of a protomer being conductive was much lower when in the intermediate conformation than in the fully open one, regardless of the conductance threshold (see Methods for details) used to distinguish between conductive and nonconductive states (Fig. 1d). Moreover, protomers in the fully open state reached higher mean

conductance ($G_m$), whereas no protomer in the intermediate state reached a mean conductance of 20 pS (Fig. 1d, inset). Additionally, we used instantaneous conductance ($G_i$), as defined by the waiting times between subsequent permeation events, to characterize the ion-conductive microstate of the pore that contributed most to ion conduction (see Methods for details). The instantaneous conductance was notably higher in the fully open than in the intermediate state of the pore (median values of 31.3 and 10.7 pS, respectively; Fig. 1e). We also conducted simulations of the cryo-EM structure of L302A nhTMEM16 captured in the intermediate state (Supplementary Table 3), which was experimentally shown to abolish lipid scrambling but to retain ion permeability[25]. The mutant demonstrated significantly lower ion-conduction probability and ion conductance than the wild type in the fully open state (Supplementary Fig. 2a, b). The values for instantaneous conductance are in a good agreement with experimentally observed single-channel conductances for the TMEM16F and fungal TMEM16 scramblases. In particular, the median instantaneous conductance for the fully open state is remarkably close to the experimentally determined single-channel conductance reported for TMEM16F as an outwardly rectifying $Cl^-$ channel (50 pS)[14]. However, the lower ion conductance demonstrated by the intermediate state and the L302A mutant is closer to the single-channel conductance (~1 pS) reported in another TMEM16F study[5], suggesting that TMEM16F can adopt different ion-conductive states similar to nhTMEM16 and that the experimental conditions might shift equilibrium between these states. Finally, the single-channel conductance reported for afTMEM16 (~300 pS)[11] is of the same order of magnitude as the conductance of a number of the fully open protomers that exhibited instantaneous conductances of above 100 pS, indicating full opening of the afTMEM16 cavity under experimental conditions.

In our simulations, the conformation of the conductive pores in the fully open state remained close to the original structure; in contrast, the conformation of the conductive pores in the intermediate state notably deviated from the tightly sealed pore of the original intermediate structure, as shown by the root mean square deviation (RMSD) analysis of the pore (Supplementary Fig. 3a, b). Importantly, in the simulations initiated from the intermediate conformation, structural similarity to the fully open state was higher in conductive than in nonconductive protomers (Supplementary Fig. 3b), suggesting a partial transition of the subunit cavity from the sealed to the open state. In particular, the transition to the conductive intermediate state was also accompanied by widening of the pore (Supplementary Fig. 3c), resulting in an increase in pore hydration (Fig. 1f). Thus, the intermediate conformation, as captured with the 6QMA structure, requires rearrangement of the pore structure to initiate ion conduction. Moreover, compared to the fully open state, the time prior the first ion permeation event was about twice as long in simulations starting from the intermediate state (Supplementary Fig. 3d) or from the L302A mutant (Supplementary Fig. 4a). Conductive pores of the L302A mutant were also wider compared to the nonconductive ones (Supplementary Fig. 4b), but retained a similar level of hydration (Supplementary Fig. 4c). Notably, additional simulations of $Ca^{2+}$-bound TMEM16F (PDB ID: 6QP6; Supplementary Table 4), which was captured in a conformation very similar to the intermediate state of nhTMEM16, indicate that its pore is strongly dehydrated (Supplementary Fig. 5a, b) with no ion conduction at voltages as high as 1 V. From these three lines of evidence (probability of conductive state, mean and instantaneous conductances, and conformational changes required to enable ion permeations), we conclude that the fully open conformation represents the main ion-conductive state of TMEM16 scramblases.

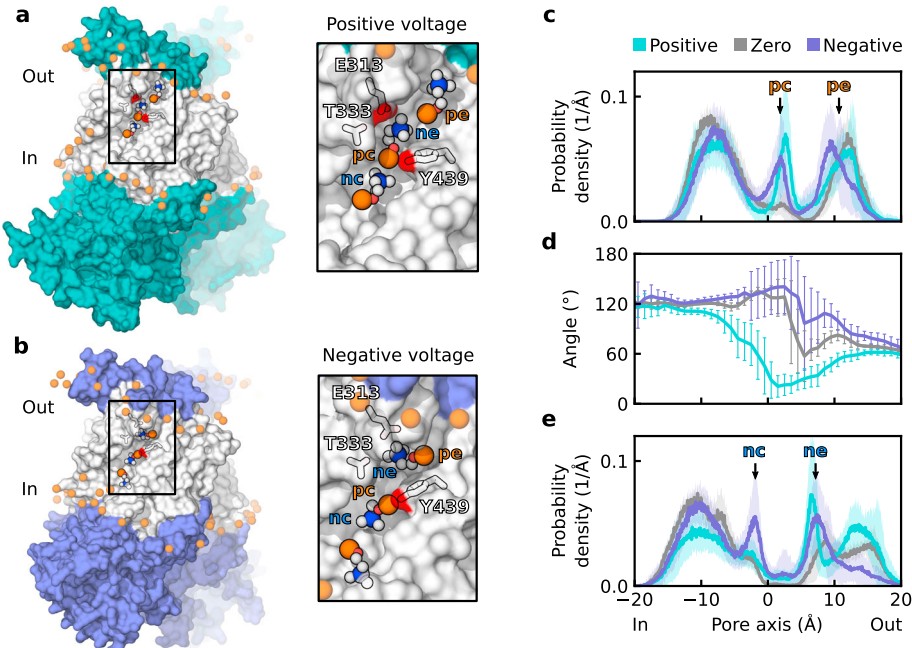

**Fig. 2 Structured within the subunit cavity, lipid headgroups demonstrate voltage polarity-dependent arrangement. a, b** Snapshots showing the upward orientation of lipid headgroups at positive voltages (**a**) and downward orientation at negative voltages (**b**), along with enlargements of the cavity region. Headgroups within the cavity and key residues that stabilize their arrangement are highlighted (phosphorus, nitrogen, oxygen, and carbon atoms are shown in orange, blue, red, and white, respectively). The transmembrane part of the protein is shown in white. **c** Local probability density distribution of POPC phosphorus atoms along the subunit cavity. Localization sites (*pc* and *pe*) are labeled. **d** Distribution of angle between a lipid headgroup and the outward membrane normal, where 0° and 180° indicate upright and downright orientations, respectively. **e** Local probability density distribution of POPC nitrogen atoms along the subunit cavity. Localization sites (*nc* and *ne*) are labeled. **c–e** Distributions were calculated with respect to the pore center. Average distributions across independent protomer simulations are shown, with error bars and shaded areas representing the standard error of mean. Data were derived from $n = 8$, $n = 19$, and $n = 21$ independent protomers at zero, positive, and negative voltages, respectively.

**Lipid headgroups are dynamic voltage-sensitive structural elements of the proteolipidic pore**. In the main ion-conductive state, the nhTMEM16 pore is characterized by a proteolipidic permeation pathway that is lined with lipid headgroups (Fig. 2a, b). To obtain structural details of the headgroup arrangement within the cavity, we calculated local probability density distributions (see Methods for details) of lipid phosphorus and nitrogen atoms within the pore along the outward membrane normal (Fig. 2c, e). The distribution peaks indicate the localization sites of POPC choline (nitrogen) and phosphate groups at the extracellular (*pe*, *ne*) and central (*pc*, *nc*) sections of the cavity (Fig. 2c, e), consistent with previous reports[13,24]. Notably, the height of the central phosphorus peak increased when a voltage was applied (Fig. 2c), suggesting that transmembrane voltages might modify lipid-scrambling rates by stabilizing headgroups within the cavity. In contrast, lipid headgroups were largely excluded from the central part of the pore in the intermediate state of wild-type and L302A nhTMEM16 (Supplementary Fig. 6a, b), consistent with its impaired scrambling functionality[25].

Interestingly, due to their inherent dipole moment, lipid headgroups within the cavity are oriented in accordance with the direction of the applied electric field (Fig. 2d). Difference in the orientation of headgroups in the presence of positive and negative voltages can be clearly seen in the central part of the cavity. More precisely, the headgroup located at the central *pc* site adopts an upward orientation at positive voltages and a downward orientation at negative voltages (Fig. 2a, b). Notably, in our simulations we observed six complete lipid-scrambling events mediated by nhTMEM16 in the fully open state. While crossing the central site, the headgroups of the translocated lipids

were oriented according to the voltage polarity (Supplementary Table 5). At positive voltage two out of three scrambled lipids were translocated from the upper to the inner leaflet of the membrane. One lipid scrambled at positive voltage and all three lipids at negative voltage were translocated in the outward direction. Although these data suggest a possible correlation of the voltage polarity and headgroup orientation with the direction of lipid translocation, further investigations (including enhanced-sampling techniques to sample lipid scrambling) would be required for a rigorous assessment of the voltage dependence of lipid-scrambling kinetics in TMEM16 proteins. Consistent with the difference in the headgroup orientation, the distribution peaks of choline nitrogen atoms differ at positive and negative voltages (Fig. 2e). In particular, the central peak (*nc*) is prominent only when a negative voltage is applied, corresponding to a downward orientation of the headgroup at the central site. At the same time, an external nitrogen peak (*ne*) is formed by a choline belonging to the headgroup located at the *pc* or *pe* site, respectively, when a negative or positive voltage is applied. Notably, at zero voltage, the centrally localized headgroup samples both upward and downward orientations (although the predominant orientation is downward; Supplementary Fig. 7a). Thus, positive voltage is the switching stimulus for the headgroup reorientation.

By measuring the contacts formed between the headgroups and cavity residues, we found that the *pe* phosphate and *nc* choline groups are rather loosely coordinated by the surrounding residues, with no residue demonstrating a greater than 20% probability of making direct contact with headgroup moieties. At the same time, groups at the *pc* and *ne* sites were notably stabilized by cavity residues (Supplementary Fig. 7b). In particular, the central lipid headgroup was held in position via

a hydrogen bond between its phosphate and Y439 (Fig. 2a, b and Supplementary Fig. 7b). Of note, a substantial role for this tyrosine in both scrambling and protein activation, has recently been demonstrated[13,25,26]. In turn, the *ne* choline group is coordinated by E313 and T333, and tryptophan substitution of the former residue was previously shown to significantly decrease the ion channel activity of nhTMEM16[26]. Despite the importance of the E313/E318/R432 triad for the scrambling activity of nhTMEM16[26], E318 and R432 seem to have less prominent roles in headgroup stabilization compared with E313: R432 has a much lower probability to coordinate the phosphate group at the *pe* site and E318 scarcely makes any contact with the headgroups (Supplementary Fig. 7b). Moreover, we found no correlation between the conformation of this triad and ion conductance of nhTMEM16 in the intermediate or the fully open state (Supplementary Fig. 8a, b). In summary, although protein residues provide a general scaffold for energetically favorable lipid interactions with the subunit cavity, the transmembrane voltage significantly influences the specific lipid headgroup arrangement.

**Lipids shape the ion pathway through the proteolipidic pore.** The proteolipidic nature of the ion-conducting pore suggests that the particular lipid headgroup arrangement directly influences the ion permeation process. To determine how lipids affect permeating ions, we investigated all ions approaching the subunit cavity region (Fig. 3a). We identified two classes of ions visiting the region: (1) permeating ions, which eventually moved through the pore across the membrane, and (2) blocked ions, which did not complete the permeation event. These classes of ions are illustrated by $Cl^-$ trajectories within a representative protomer at a negative voltage (Fig. 3a).

A few dwelling sites along the ion permeation pathway were identified based on the distribution of permeating ions along the cavity (Fig. 3b, c). Notably, maximum dwell time of the ions at the sites was by one order of magnitude shorter than that of the lipid headgroups (Supplementary Fig. 9a, b), indicating a clear time-scale separation between ion permeation and lipid scrambling. An accumulation site for $Na^+$ ions at the intracellular side of the pore ($Na_i$) (Fig. 3b) mainly results from the attraction of $Na^+$ to D367 at the intracellular entrance to the cavity (Supplementary Fig. 10a, c). In turn, an extracellular $Na^+$ localization site ($Na_e$) is formed by the *pe* phosphate and the aforementioned E313 (Fig. 3b and Supplementary Fig. 10a, c), additionally emphasizing the importance of this glutamate residue for ion conduction. Finally, the vestibular $Na^+$ accumulation site ($Na_v$) is only formed when a positive voltage is applied (Fig. 3b): $Na^+$ ions are then attracted by the central phosphate group (Supplementary Fig. 10a). At negative voltages, the *nc* choline group screens the central phosphate group (Fig. 2e) and prevents $Na^+$ accumulation at this site. Since lipid headgroups were excluded from the pore center in the intermediate state (Supplementary Fig. 6a), the $Na_v$ site was not formed there (Supplementary Fig. 11a).

The position of the vestibular peak in the permeating $Cl^-$ distribution ($Cl_v$) also depends on the electric field orientation, such that at negative voltages the effluxing anions are stabilized closer to the pore center (Fig. 3c) by the *nc* choline group (Supplementary Fig. 10b). However, the general asymmetry of the cavity could also play a role in the peak shift, since it was also observed in the intermediate state of nhTMEM16 (Supplementary Fig. 11b). At the same time, the peak corresponding to the extracellular $Cl^-$ site ($Cl_e$) shifts in a polarity-dependent manner (Fig. 3c): it moves closer to the pore center when a negative voltage is applied. The shift results from a change in permeation

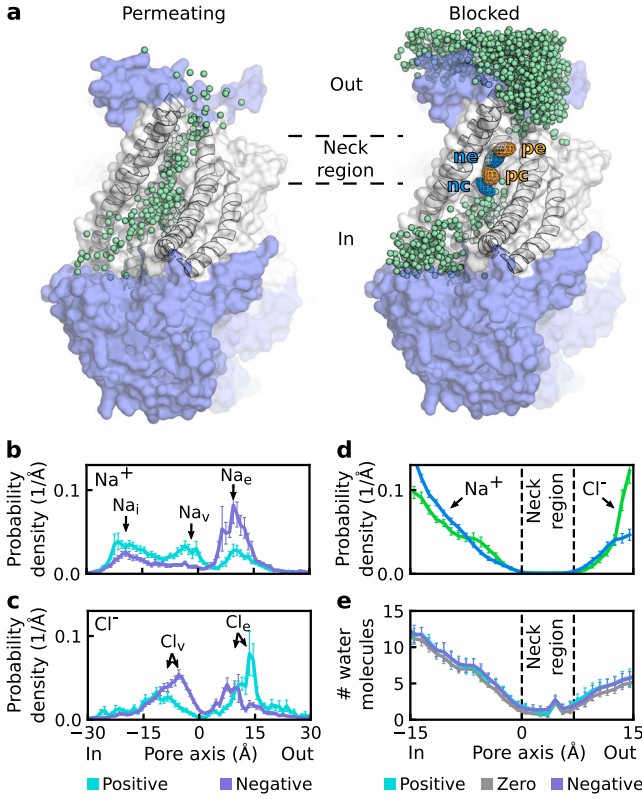

**Fig. 3 Ion-permeation pathway and neck region of the proteolipidic pore. a** Snapshots showing the trajectories of permeating (left) and blocked (right) $Cl^-$ ions visiting the nhTMEM16 pore region in a representative system with the protein at a negative voltage. Each green sphere represents a particular position of a $Cl^-$ ion in a single frame of the simulation. The neck region (indicated by dashed lines) is visited only by permeating ions. Contoured at $20\sigma$, meshes represent the density of the four phosphorus (orange) and nitrogen (blue) atoms of POPC located nearest to the pore center. The localization sites of headgroup moieties (*nc*, *pc*, *ne*, and *pe*) within the cavity are indicated. **b, c** Probability density distributions of permeating $Na^+$ (**b**) and $Cl^-$ (**c**) ions along the pore. Ion-localization sites ($Na_i$, $Na_v$, $Na_e$, $Cl_v$, $Cl_e$) are indicated. In **b** data were derived from $n = 18$ and $n = 13$ independent protomers at positive and negative voltages, respectively. In **c** data were derived from $n = 17$ and $n = 22$ independent protomers at positive and negative voltages, respectively. **d** Probability density distributions of blocked $Cl^-$ and $Na^+$ ions along the pore. Data were derived from $n = 44$ independent protomers. **e** Hydration profile of the pore, represented by the number of water molecules in 1 Å sections along the pore axis. Data were derived from $n = 8$, $n = 19$, and $n = 21$ independent protomers at zero, positive, and negative voltages, respectively. **b–e** Error bars represent the standard error of mean.

direction that causes $Cl^-$ ions to accumulate either below or above R432 and the *ne* choline group, which jointly coordinate anions at this site (Supplementary Fig. 10b, c). In conclusion, although protein residues (in particular E313 and R432) play the major role in shaping the extracellular part of the ion permeation pathway, lipid headgroups and their voltage polarity-dependent orientation strongly affect ions in the intracellular vestibule of the pore.

**Lipid headgroups control the permeability state of the pore.** In contrast to permeating ions, blocked ions showed a clear gap in distribution, corresponding to the neck region of the pore (Fig. 3a). The notable impact of lipid headgroups on permeating ions suggests that accessibility of the neck region to ions and,

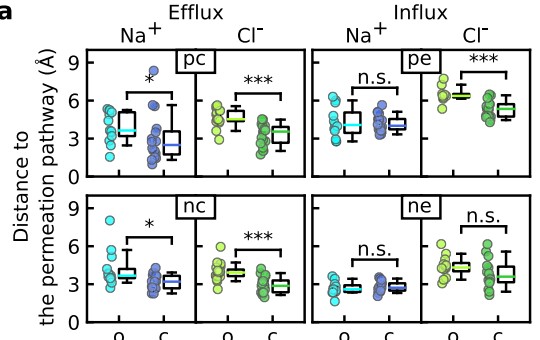
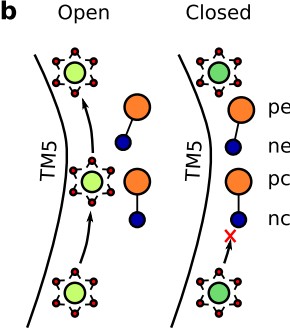

**Fig. 4 Lipid headgroups control the permeability state of the pore. a** The minimum in-plane distance between the ion permeation pathway and the nitrogen or phosphorus atoms of POPC headgroups was measured at the indicated headgroup localization sites of the open (o) and closed (c) states of the pore in the fully open conformation. The central (pc and nc) and extracellular (pe and ne) sites were used for the analysis of the efflux (left) and influx (right), respectively. Each data point represents an independent protomer, and boxplots are defined as follows: the middle line is the median, the lower and upper hinges correspond to the first and third quartiles, whiskers show the 5th and 95th percentiles. Significance was evaluated with the Mann–Whitney test, one-sided: $p > 0.05$ (n.s.), $*p < 0.05$, $***p < 0.001$. Number of data points and $p$ values ($n_o$, $n_c$, $p$): top row—(13, 16, 0.034), (17, 18, 0.00003), (12, 18, 0.5), (9, 17, 0.0009); and bottom row—(13, 17, 0.01), (20, 19, 0.00004), (11, 13, 0.3), (12, 15, 0.07). **b** Schematic representation of the efflux of fully hydrated (water oxygens shown in red) Cl⁻ ions (green) through the pore in the open state and of the blockage of permeation by lipid headgroups (orange and blue) in the central part of the pore. TM5 transmembrane domain 5.

therefore, the permeability state of the pore could also be affected by lipid headgroups. To test this, we first quantitatively characterized the neck region. For this, we defined the neck region as the section of the pore with a negligible probability density of blocked ions (Fig. 3d). The region extends for a distance of about 7 Å from the pore center to its extracellular entrance, corresponding to the least hydrated part of the cavity (Fig. 3e). Despite this, we noted that ions retain their first hydration shell upon crossing the neck region (Supplementary Fig. 12a). Notably, the neck region overlaps with the localization sites of lipid headgroups (Fig. 3a).

To test whether lipid headgroups affect the permeability state of the pore, we measured the distances from headgroup moieties to the ion permeation pathway (see Methods for details) and compared these between the open (ions are permeating) and closed (ions are blocked) states of the pore in the fully open conformation (Fig. 4a). As entry to the neck region was sufficient for most ions to permeate the pore, we measured the effects of extracellular (pe and ne) and central (pc and nc) headgroup moieties on ion influx and efflux, respectively. When the central part of the pore was open and ion efflux was permitted, both the choline and phosphate groups of the central lipid headgroup were significantly further from the ion permeation pathway than when the pore was in the closed state (Fig. 4a). The blockage is likely to be mediated by steric hindrance as, regardless of their polarity, both headgroup moieties interfere with Cl⁻ and Na⁺ efflux.

Ion influx was less affected by lipid headgroups: only the extracellular phosphate group (pe) significantly affected Cl⁻ influx through the pore (Fig. 4a). Notably, blockage of Cl⁻ influx (unlike blockage of Cl⁻ efflux and Na⁺ influx and efflux) additionally correlated with changes in the pore structure, as indicated by RMSD analysis (Supplementary Fig. 12b). In summary, ion efflux is controlled by lipid headgroups that can sterically block access to the neck region and the ion permeation pathway (Fig. 4b), whereas ion influx is regulated by both lipids and protein conformational dynamics.

**Anionic lipids reverse the ion selectivity of nhTMEM16.** Lipid headgroups interact with permeating ions and control the permeability state of the pore, and are, therefore, expected to shape the energetics of ions along the permeation pathway. Since differences in the energetics of the different ions define the ion

selectivity of the pore, lipids may have a profound impact on ion selectivity. To analyze the ion selectivity of nhTMEM16, we first sorted the simulated protomers into five selectivity classes based on their Na⁺-to-Cl⁻ permeability ratio ($P_{Na}/P_{Cl}$; see Methods for details); the permeability ratios weighted over the protomers are shown in Supplementary Table 6. Selectivity of the proteolipidic pore was found to depend on voltage polarity: protomers at negative voltages were moderately selective for Cl⁻ and those at positive voltages had nonselective currents (Fig. 5a). Structurally, the difference in selectivity stems from the asymmetry of the subunit cavity, in which the intracellular vestibule serves as a priming site for effluxing ions. Therefore, upon switching from negative to positive voltages, the priming site for Cl⁻ is lost but a priming site is created for Na⁺, leading to a shift in the ion selectivity of nhTMEM16 toward the nonselective state.

When we changed the lipid composition of the membrane to a 1:1 mixture of POPC and negatively charged 1-palmitoyl-2-oleyl-phosphatidylserine (POPS), keeping a NaCl concentration of 250 mM, we observed a drastic change in ion selectivity, such that the proteolipidic pore of fully open nhTMEM16 became effectively Na⁺ selective (Fig. 5b; Supplementary Table 6). Notably, the same effect was evident in a 3:1 mixture of neutral 1-palmitoyl-2-oleyl-phosphatidylethanolamine (POPE) and negatively charged 1-palmitoyl-2-oleyl-phosphatidylglycerol (POPG) (Supplementary Fig. 13a; Supplementary Table 6). Sequence comparison of nhTMEM16 with the fungal homolog afTMEM16 revealed a charge change of two residues (N435 and D367 in nhTMEM16 are substituted by K428 and K359 in afTMEM16), which are in contact with permeating ions in nhTMEM16 (Supplementary Fig. 10a, b). The positive charge brought by these two basic residues predicts a reduced $P_{Na}/P_{Cl}$ in afTMEM16 compared with nhTMEM16. Indeed, the experimental value of $P_K/P_{Cl} \sim 1.5$, determined for afTMEM16 reconstituted in a POPE:POPG membrane[11], is lower than $P_{Na}/P_{Cl} = 8.7$ of nhTMEM16, which we observed in the simulations conducted in a membrane of the same composition (Supplementary Table 6). Despite the drastic change in the ion selectivity, ion conductance (Supplementary Fig. 13b) and the general ion-conduction mechanism remain similar to those in the pure POPC membrane. In particular, the ion-permeation pathway is lined by lipid headgroups (Supplementary Fig. 14a, b), with ions dwelling at the vestibular and extracellular sites (Supplementary Fig. 15a–d). Notably, unlike the POPC and POPE headgroups in the mixed

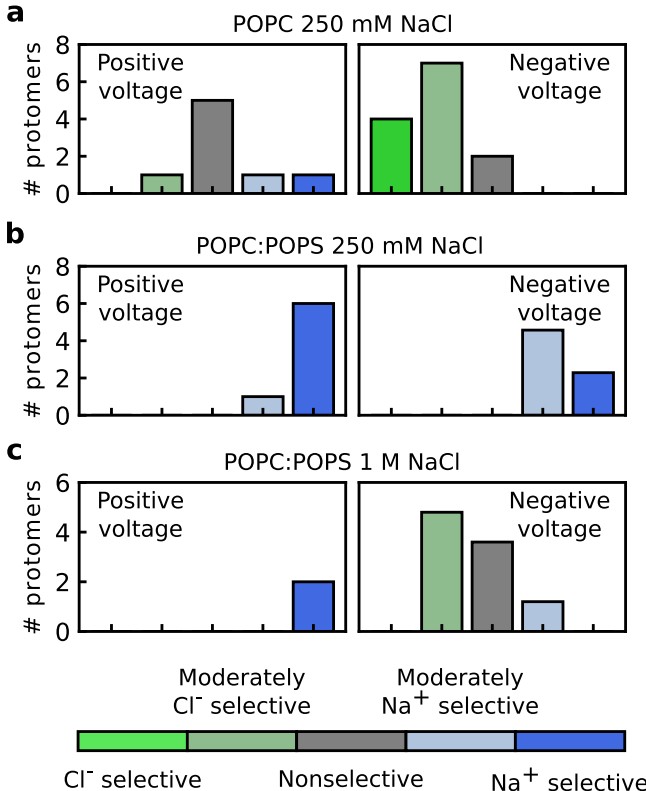

**Fig. 5 Ion selectivity of nhTMEM16 depends on voltage polarity, membrane lipid composition, and salt concentration. a–c** The number of nhTMEM16 protomers in a certain selectivity class in a pure POPC membrane (**a**) and a POPC:POPS (1:1) membrane at either 250 mM NaCl (**b**) or 1 M NaCl (**c**) is shown for positive (left) and negative (right) voltages. A protomer was considered to be nonselective if its permeability ratio was <2 (gray), moderately selective if it was 2–10 (pale green and pale blue), and selective if it was ≥10 (green and blue). Results are shown only for protomers that had undergone at least five ion permeation events by the end of the simulation.

membranes, which have a localization pattern similar to that of the pure POPC membrane, the headgroups of anionic lipids rarely visit the central part of the pore (Supplementary Fig. 14a, b). Such depletion suggests slower kinetics of scrambling for phosphatidylserine compared with phosphatidylethanolamine lipids, in full agreement with experimental observations[11].

When we increased the NaCl concentration from 250 mM to 1 M, the ion selectivity of nhTMEM16 embedded in POPC:POPS membrane reversed to partially restored preference for Cl⁻ in pores under negative voltages (Fig. 5c). The increased salt concentration seems to result in ion selectivity that is intermediate between those of neutral and anionic membranes at the lower salt concentration. Thus, as the protein demonstrates nonselective current at positive voltages in the POPC membrane, its ion selectivity is shifted towards Na⁺ in the POPC:POPS membrane even at the higher salt concentration. Moreover, when we reduced the NaCl concentration to 200 mM and added Ca²⁺ ions to the solution to reach bulk concentration of ~1 mM, we also observed a similar reduction in $P_{Na}/P_{Cl}$ (Supplementary Fig. 16; Supplementary Tables 6, 7). Consistent with our observations, a shift to more Cl⁻-selective currents in increased salt concentration was recently measured for TMEM16F[41]. Taken together, the ability of anionic lipids to alter ion selectivity, the irrelevance of the particular chemical structure for the selectivity change, the exclusion of anionic lipids from the central part of the

subunit cavity, and the partial reversal of ion selectivity at higher salt concentrations indicate that membrane surface charge at the pore entrances is an important modulator of ion selectivity in TMEM16 lipid scramblases.

**The TMEM16K proteolipidic pore is highly structured**. To assess whether mammalian members of the TMEM16 family share the ion conduction mechanism identified in nhTMEM16, we carried out a series of simulations of the human lipid scramblase TMEM16K using the recently resolved X-ray structure of the protein in its fully open state[10]. The protein was embedded in a POPC membrane and subjected to various voltages from 250 to 650 mV, with a bulk NaCl concentration of ~250 mM (see Supplementary Table 8). The simulations confirmed that the proteolipidic-pore model of ion conduction is generally applicable to TMEM16 lipid scramblases (Fig. 6a, b). However, lipid headgroups within the TMEM16K subunit cavity are more ordered compared with nhTMEM16 (Fig. 6a, b), leading to more peaks in the phosphorus and nitrogen distributions (Fig. 6c, d). The increased headgroup order stems from an extended network of interactions that stabilize headgroups (Supplementary Fig. 17). In particular, the intracellular $p1$, $p2$, and $n1$ sites are formed by R375, N444, and D496, respectively, while the extracellular $n4$ and $p4$ groups are stabilized by Q431 and E340 (Fig. 6a, b and Supplementary Fig. 17), which are homologs of the N435 and E313 residues that coordinate the $pe$ and $ne$ groups in nhTMEM16. Surprisingly, replacement of the bulky Y439 with the short T435 did not lead to loss of the central localization site ($p3$) due to an accompanying substitution of A444 by S440, which stabilizes the $p3$ phosphate group and contributes to coordination of the $n2$ choline group (Fig. 6a, b and Supplementary Fig. 17). In turn, T435 as well as S363 is involved in coordination of the choline group at the extracellular $n3$ site. Finally, neither the $n0$ nor $n5$ choline groups form stable contacts with any of the cavity-lining residues; instead, they drive from a particular headgroup orientation (described below).

The orientation of lipid headgroups in the TMEM16K cavity demonstrates the same sensitivity to voltage polarity as in the nhTMEM16 cavity (Fig. 6e). Sensitivity to voltage polarity is especially pronounced in the weakly hydrated neck region of the TMEM16K pore (Fig. 6f, g), which is about 10 Å longer than in nhTMEM16. The $n0$ peak depends on the headgroup orientation at the $p1$ site and, therefore, on voltage polarity (Fig. 6d): it is much more prominent at negative voltages similar to the $nc$ peak in nhTMEM16 (Fig. 2e). Relatedly, at negative voltages the choline groups at the $n1$ and $n2$ sites are part of headgroups whose phosphates localize to the $p2$ and $p3$ sites, respectively, whereas at positive voltages the same choline sites are occupied by headgroups whose phosphates localize to the $p1$ and $p2$ sites (Fig. 6a, b). Overall, our analysis confirmed the general properties of the proteolipidic pore as inferred from nhTMEM16 simulations, including the ordering of lipid headgroups and the polarity change-induced exchange of choline groups between localization sites.

**TMEM16K mediates cation-selective currents**. Despite the similarity of the nhTMEM16 and TMEM16K proteolipidic pore structures, we observed a striking difference in ion selectivity between the fungal and mammalian homologs. In contrast to nhTMEM16, TMEM16K demonstrated a strong preference to conduct cations over anions: all simulated protomers were more than twice as permeable to Na⁺ than to Cl⁻ (Fig. 7a). To uncover the molecular basis of this difference, we scrutinized the amino-acid sequences of the pore-lining regions. We found that charged residues of transmembrane domain (TM) 3 and TM6 line the

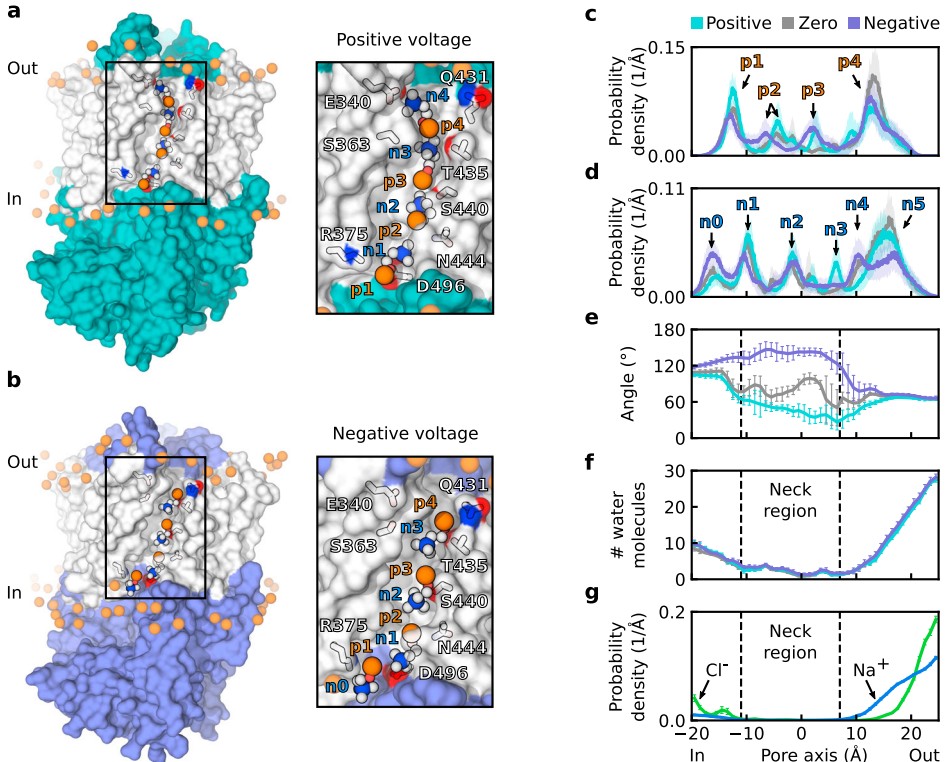

**Fig. 6 Structure and properties of the TMEM16K proteolipidic pore. a, b** Snapshots showing the arrangement of lipid headgroups within the subunit cavity of TMEM16K at positive (**a**) and negative (**b**) voltages. Lipid headgroups within the cavity and the key residues coordinating them are highlighted, with phosphorus, nitrogen, oxygen, and carbon atoms shown in orange, blue, red, and white, respectively. The part of the protein within the hydrophobic core of the membrane is shown in white. **c, d** Local probability density distributions of the phosphorus (**c**) and nitrogen (**d**) atoms of POPC along the subunit cavity. Localization sites (n0, p1, n1, p2, n2, p3, n3, p4, n4, and n5) are labeled. **e** Distribution of angles between a lipid headgroup and the outward membrane normal, where 0° and 180° indicate the upright and downright orientations, respectively. **f** Hydration profile of the pore, represented by the number of water molecules in 1-Å sections along the pore axis. **c–f** Data were derived from $n = 4$, $n = 14$, and $n = 14$ independent protomers at zero, positive, and negative voltages, respectively. **g** Probability density distributions of blocked $Cl^-$ and $Na^+$ ions along the pore. Data were derived from $n = 28$ independent protomers. **c–g** The distributions were calculated relative to the pore center. The average distributions over the independent protomer simulations are shown, with error bars and shaded areas representing the standard error of mean.

extracellular entrance to the cavity, and charged residues of TM4 and TM7 line the intracellular entrance (Fig. 7b). The conserved acidic residues, which form the $Ca^{2+}$-binding site (D503 and E506 in nhTMEM16, and D497 and E500 in TMEM16K), do not interact directly with permeating ions (Supplementary Fig. 10a, b; Supplementary Fig. 18c, d), and are thus not considered here. We also note that properties of the extracellular entrance to the pore in other isoforms (such as TMEM16F) could additionally be affected by the ectodomain.

While all three charged residues of the nhTMEM16 cavity point toward the inside of the groove, some of the TMEM16K residues (especially the basic ones) point toward the outside (Fig. 7c). In particular, the side chains of both K427 and R430 are oriented such that they cannot neutralize the negative charges of D425 and E340 that are directed toward the extracellular entrance. The same region is effectively neutral in nhTMEM16 due to the close positioning of R432 to E313. At the intracellular entrance, R505 creates an anion-attracting environment in nhTMEM16, whereas its acidic homolog D496 is effectively neutralized by R378 in TMEM16K. The human scramblase has three charged residues (R378, R375, and E371) in the pore-lining stretch of TM4, whereas nhTMEM16 contains only neutral residues in the same region. In the crystal structure, R375 points out of the cavity; however, during the simulations R375 reoriented its side chain toward the cavity interior, where it coordinated the p1 phosphate (Fig. 6a, b) and thus could not

effectively screen the negatively charged E371. We conclude that the positioning and orientation of the charged residues form the basis for the notable difference in pore electrostatics between the homologs.

Due to the low $Cl^-$ conductance of TMEM16K, we focused our further analysis on $Na^+$ permeation. Intracellular ($Na_i$), central ($Na_c$), and extracellular ($Na_e$) $Na^+$ localization sites were defined based on the distribution of the permeating ions along the pore (Supplementary Fig. 18a). All three sites are partly formed by the lipid phosphate groups and demonstrate a dependence on voltage polarity (Supplementary Fig. 18a). In particular, the $Na_i$ and $Na_e$ sites are prominent only at positive and negative voltages, respectively, when there is no adjacent choline group (n0 and n3) (Supplementary Fig. 18a and Fig. 6d). Thus, $Na^+$ ions can be freely coordinated by a corresponding phosphate group in the absence of an interfering choline group. In turn, the $Na_c$ site demonstrates a voltage polarity-dependent shift, following the location of the p2 site (Supplementary Fig. 18a and Fig. 6c). Importantly, $Na^+$ is further stabilized within the cavity by several residues, including E340, D496, and E371 (Supplementary Fig. 18c, d), which highlights the importance of these residues in cation selectivity of the pore. Similar to nhTMEM16, $Na^+$ ions retain their hydration shell while passing through the neck region of TMEM16K (Supplementary Fig. 18b), suggesting that the ions bind rather loosely to the site-forming residues. Notably, $Na^+$ ions are stabilized at their accumulation sites by TM3, TM4, and TM5

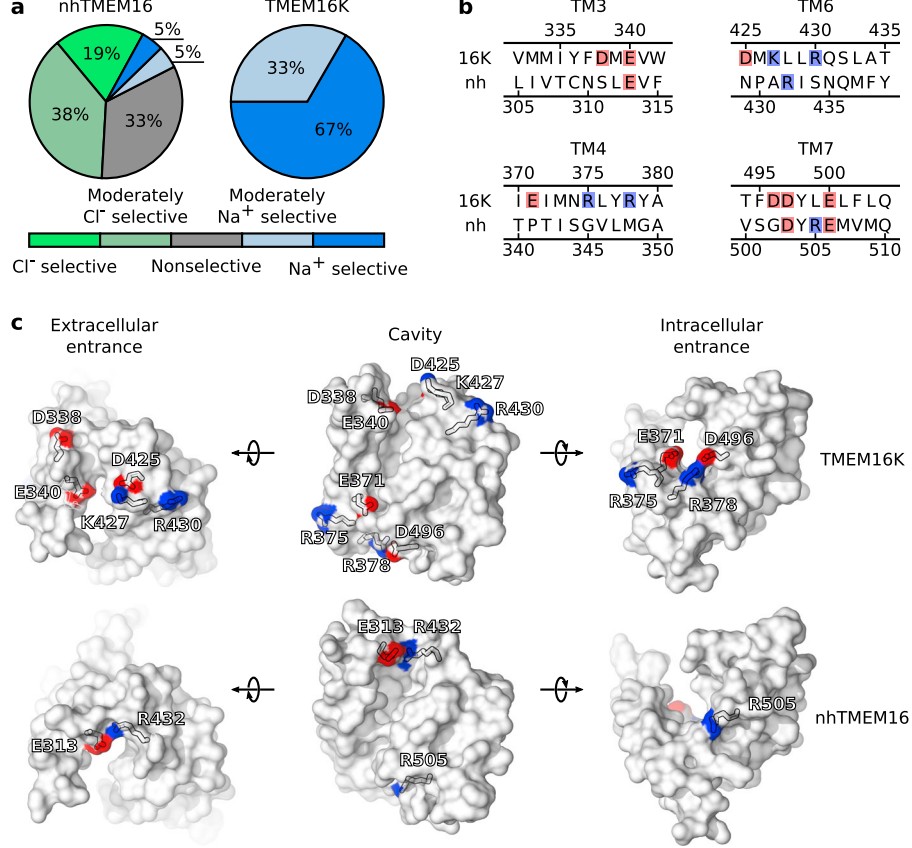

**Fig. 7 Structural determinants of TMEM16K cation selectivity. a** Percentages of simulated protomers that demonstrate various levels of ion selectivity in nhTMEM16 and TMEM16K simulations. Data on ion selectivity for both positive and negative voltages were combined. **b** Alignment of nhTMEM16 and TMEM16K sequences at the regions forming the intracellular (TM3 and TM6) and extracellular (TM4 and TM7) entrances of the pore and containing negatively and positively charged residues highlighted in red and blue, respectively. TM transmembrane domain. **c** Snapshots showing the distribution of charged residues within the subunit cavities of nhTMEM16 and TMEM16K. Oxygen and nitrogen atoms of the side chains are colored red and blue, respectively.

residues (Supplementary Fig. 18c, d). In contrast, lipid headgroups are mainly coordinated by TM6 residues (Fig. 6a, b). Together with the absence of a blockage effect by headgroups on ion permeation (Supplementary Fig. 19), this indicates a looser coupling between lipid scrambling and ion conduction in TMEM16K compared with nhTMEM16. In summary, ions permeating through a proteolipidic pore can be affected to varying degrees by the pore-lining lipid headgroups, depending on the particular amino-acid composition of the scramblase cavity.

## Discussion

The TMEM16 protein family comprises $Ca^{2+}$-activated $Cl^-$ channels and lipid scramblases. In addition to transporting lipids, TMEM16 lipid scramblases mediate ion currents[5–8,10–14,25,26], and are implicated in various physiological processes[5,14,17]. In this study, we used extensive atomistic MD simulations of fungal nhTMEM16, human TMEM16K, and murine TMEM16F to gain atomic-level insights into the ion conduction mechanism of TMEM16 lipid scramblases. We found that the fully open conformation represents the main conductive state of nhTMEM16. In this state, nhTMEM16 conducts ions through a structured but still dynamic proteolipidic pore, which is lined by lipid headgroups and the residues of the subunit cavity. Oriented in a voltage polarity-sensitive way, lipid headgroups shape the ion permeation pathway and directly control permeability state of the nhTMEM16 pore. We also discovered that the ion selectivity of

the nhTMEM16 pore is drastically affected by the membrane lipid composition in a salt concentration-dependent manner. Next, we demonstrated that the proteolipidic-pore conduction mechanism of nhTMEM16 also applies to human TMEM16 lipid scramblases. By applying CompEL simulations to TMEM16K, we circumvented the difficulties of studying this isoform experimentally due to its localization in the endoplasmic reticulum (ER)[10] to find that it forms a cation-selective pore. Finally, we elucidated the structural determinants of cation selectivity in TMEM16K and of the reduced effect of lipids on TMEM16K-mediated ion conduction.

Our simulations indicate that the $Ca^{2+}$-bound intermediate conformation of nhTMEM16, as obtained by cryo-EM[33], requires conformational rearrangements in the pore and a subsequent increase in pore hydration to initiate ion conduction (Fig. 1, Supplementary Fig. 3). However, the pore of the conductive protomers in the intermediate state is not wide enough to accommodate lipid headgroups, which are excluded from the central part of the pore (Supplementary Fig. 6a). Similarly, lipid headgroups rarely visit the central part of the pore of the nhTMEM16 L302A mutant (Supplementary Fig. 6b). Notably, the ion-conductance level of the mutant and the wild-type protein in the intermediate state is considerably reduced compared with the fully open conformation of nhTMEM16 (Fig. 1d, e, Supplementary Fig. 2). However, even such low conductance would be sufficient for detection in liposome flux assays, in agreement with the recent study on the L302A mutant[25]. Given that in the

simulations pore dehydration prevented ion conduction by the $Ca^{2+}$-bound TMEM16F (Supplementary Fig. 5), we conclude that $Ca^{2+}$ binding alone is insufficient to activate ion conduction and that the complete opening of the subunit cavity is required for TMEM16 lipid scramblases to transition to the main ion-conductive state, represented by the proteolipidic pore. This conclusion is consistent with the delayed but concomitant activation of both ion conduction and lipid scrambling in TMEM16F[13]. Finally, the resolved structure of the pore of the TMEM16A ion channel[28,29] resembles those of nhTMEM16 and TMEM16F in their intermediate conformations[22,33]. Given that in our simulations single-channel conductance of nhTMEM16 in the intermediate state (~10 pS) was comparable to that of TMEM16A (~3–8 pS[3,40]), it is tempting to speculate that ion conduction in the intermediate state resembles that of bona fide channels of the family. By contrast, fully active TMEM16 lipid scramblases conduct ions through the high-conductance proteolipidic pore.

We then showed that in both fungal and human scramblases the arrangement of lipid headgroups forming part of the proteolipidic pore is influenced by the applied electric field, as well as by a network of interactions with the pore-lining residues (Figs. 2 and 6). In particular, Y439 in nhTMEM16 and S440 in TMEM16K coordinate the lipid phosphate groups in the central part of the cavity and an extracellular glutamate (E313 in nhTMEM16 and E340 in TMEM16K) additionally stabilizes the polarity-dependent orientation of lipid headgroups along the pore. Notably, through altering the headgroup orientation, voltage polarity affects the position of the ion-dwelling sites within the scramblase cavities (Fig. 3). In nhTMEM16, the headgroup orientation also influences the ion selectivity of the pore by inducing the conversion from essentially nonselectivity to moderate anion selectivity upon switching from a positive to a negative voltage (Fig. 5). Finally, lipid headgroups that are located at the intracellular entrance to the low-hydrated neck region and coordinated by Y439 sterically control the ion permeability of the nhTMEM16 pore (Fig. 4). A similar mechanism can be expected in TMEM16F, where the bulky Y563 residue points into the cavity and forms an inner activation gate for scrambling[31]. In all, our data prove against the "alternating pore-cavity" model, support the proteolipidic pore as the main ion conduction mechanism in TMEM16 lipid scramblases, and highlight the prominent role of lipids in ion conduction by nhTMEM16.

Strikingly, we observed a drastic change from a moderately $Cl^-$-selective to a $Na^+$-selective state in the nhTMEM16 pore upon changing from a neutral to an anionic lipid membrane (Fig. 5). This finding supports the hypothesis that the ion conduction properties of TMEM16 lipid scramblases would depend on the membrane lipid composition if ion permeation were to take place via the proteolipidic-pore mechanism[42]. Anionic lipids were excluded from the central part of the cavity and changed the ion selectivity by bringing negative charges to the pore entrances (Supplementary Fig. 14a, b). Since in the intermediate state lipid headgroups also populated the pore entrances (Supplementary Fig. 6a), we suggest that lipids could affect ion selectivity in the intermediate state as well. By increasing the salt concentration and thus reducing this negative charge, we could partially reverse the change in ion selectivity (Fig. 5, Supplementary Fig. 16). These results therefore provide the molecular basis for the recently reported dependence of TMEM16F ion selectivity on monovalent and divalent cation concentration[41]. Based on the similar dependence on salt concentration and the similar charged-residue composition of the nhTMEM16 and TMEM16F cavities, we propose that differences in lipid and ion composition in the local protein environment could result in the highly variable ion selectivity demonstrated by TMEM16F. In particular, the

cation selectivity observed in inside-out patch-clamp experiments on TMEM16F[5,21,22] could result from the anion lipid-rich environment native to the protein. This environment could change during the delayed activation of TMEM16F in whole-cell patch-clamp experiments[6,7,14], leading to anion-selective currents. Finally, the involvement of anionic lipids in the conduction process could explain the transition of the TMEM16A scrambling-competent mutant from anion selectivity to cation selectivity with the onset of lipid scrambling[13]. Although awaiting experimental validation, the mechanisms of lipid-dependent control of ion selectivity we have revealed offer one more perspective into how the cell membrane directly modulates the functional properties of TMEM16 lipid scramblases.

Although the TMEM16K simulations confirmed the voltage polarity-dependent structure of the proteolipidic pore (Fig. 6), they revealed a weaker coupling between lipid scrambling and ion conduction in TMEM16K than in nhTMEM16 owing to differences in the amino-acid composition of the cavities. In particular, lipid headgroups do not significantly affect either the permeability state of the pore or ion selectivity in TMEM16K. The lack of effect on ion selectivity derives from the intrinsic cation selectivity of the protein, as defined by the position and orientation of charged residues lining its subunit cavity (Fig. 7). Not yet achieved by experimental electrophysiological recordings, our detailed description of TMEM16K ion selectivity suggests possible functions for this ubiquitously expressed protein. The deranged $Ca^{2+}$ signaling leading to spinocerebellar ataxia in patients with mutations in TMEM16K[43] could, at least partly, result from impaired function of TMEM16K as ER-localized cation-selective ion channel. Whereas the wild-type protein would normally compensate for the loss of intra-ER charge, reduced cation influx in these mutants would be expected to attenuate $Ca^{2+}$ release from the ER. Although yet speculative, the proposed scenario would suggest an important physiological role of TMEM16K as a cation channel.

So far, the reported effects of lipids on the conduction properties of ion channels have largely been limited to indirect mechanisms, mediated either allosterically or via changes in membrane properties[44,45]. The direct modulatory effects of lipids on ion conduction reported here are likely to be relevant to the growing range of ion channels in which lipid headgroups are found to be directly implicated in the ion-permeation pathway, including TMEM16 ion channels[28,29] and P2X ionotropic receptors[46,47]. In conclusion, we expect that the molecular details of scramblase-mediated ion conduction we have uncovered, together with the proposed functional and physiological implications, will provide an important foundation for further investigations into ion conduction in ion channels with lipid-lined pores.

## Methods

**Protein models.** The fully open state of nhTMEM16 was modeled based on the X-ray structure (PDB ID: 4WIS)[23] with a final residue range of 18–722 and the cryo-EM structure (PDB ID: 6QMA)[33] was used to build a model of the intermediate state with a final residue range of 13–722. The L302A mutant of nhTMEM16 was modeled based on the cryo-EM structure (PDB ID: 6OY3)[25] with a final residue range of 13–722. The parts of the intracellular domain unresolved in the mutant structure were modeled based on the 6QMA structure. The fully open state of TMEM16K was modeled based on the X-ray structure (PDB ID: 5OC9)[10] with a final residue range of 14–642. The $Ca^{2+}$-bound TMEM16F was modeled based on the cryo-EM structure (PDBID: 6QP6)[22] with final residue range 41–879. In all protein models, the N- and C-termini were capped with acetyl and N-methyl, respectively. All $Ca^{2+}$ ions present in the original structures of nhTMEM16 and TMEM16F were preserved, whereas the luminally bound $Ca^{2+}$ ion present in TMEM16K was removed after system equilibration because it almost immediately dissociated when position restraints were removed. Missing residues that were unresolved in the original structures were restored using Modeller version 9.18[48]. The standard protonation state at neutral pH was assigned to all residues, except for protonated H611 in nhTMEM16 and H844 in TMEM16F, which are located at

the dimer interface and form salt bridges with residues E610 and E803 of the other protomer in nhTMEM16 and TMEM16F, respectively.

**Simulation parameters**. All simulations were conducted using the GROMACS software package[49] versions 2016, 2018, and 2019. CHARMM36 force field parameters were used for lipids[37] and CHARMM36m force field parameters for proteins[38]. Ions were described using default CHARMM parameters and the CHARMM TIP3P model was used for water molecules. Integration time step of 2 fs was used and all hydrogen-involving bonds were constrained with LINCS[50]. Van der Waals interactions were calculated with the Lennard-Jones potential and a cutoff radius of 1.2 nm, with forces smoothly switched to zero in the range of 1.0–1.2 nm and no dispersion correction applied. Electrostatic interactions were calculated by the particle mesh Ewald method[51], with a real-space cutoff distance of 1.2 nm. All simulations were done in isothermal-isobaric ensemble, with the temperature set to 310 K using the v-rescale thermostat[52] and a time constant of 0.5 ps. The thermostat was applied separately to a protein with bound $Ca^{2+}$ ions, a lipid bilayer, and a water solution containing ions. The same groups were used for the removal of center-of-mass linear motion. A pressure of 1 bar was imposed using either a Berendsen[53] or Parrinello-Rahman[54] barostat in a semi-isotropic manner with a time constant of 5 ps.

**System preparation**. Protein models were embedded in the equilibrated lipid membranes using the g_membed functionality[55] in GROMACS. The initial protein orientation within the membrane was guided by the corresponding structure from the Orientations of Proteins in Membranes database[56]. Pure POPC membranes were composed of neutral POPC molecules. Negatively charged POPE:POPG membranes were composed of a 3:1 ratio of neutral POPE and anionic POPG. Negatively charged POPC:POPS membrane contained a 1:1 ratio of POPC and anionic POPS. The lipids were symmetrically distributed between the leaflets of the mixed membranes to model the equilibrated lipid distribution in presence of an activated lipid scramblase. Chloride and sodium ions were added to reach a symmetric bulk salt concentration of (unless otherwise stated) 250 mM. All bilayers were equilibrated in the aqueous NaCl solution for at least 500 ns prior to protein insertion.

**Equilibration**. The system comprising a lipid bilayer and embedded protein was equilibrated in four steps. First, water and ions were equilibrated for 4 ns with all $Ca^{2+}$ ions and protein and lipid heavy atoms restrained by a harmonic potential with a force constant of 1000 kJ $mol^{-1}$ $nm^{-2}$. The lipid heavy atoms were restrained only in the z dimension. The water oxygens resolved in the crystal structure of TMEM16K were also restrained. Second, restraints on the lipid heavy atoms (except for phosphorus) were removed and lipids were allowed to equilibrate in the xy plane around the protein for 50–200 ns. Third, restraints on the phosphorus atoms and on protein loops that were missing in the original structures were released so that the latter could relax for another 50 ns. Finally, side chains and $Ca^{2+}$ ions were allowed to equilibrate for 10 ns, with restraints only on the backbone atoms of non-loop protein regions. In TMEM16K systems, restraints on the water oxygens present in the crystal structure were also released.

**Computational electrophysiology**. The CompEL method[35,36] was used to apply sustained electrochemical potential gradients across the membranes. Briefly, a CompEL system contains two lipid bilayers separating two compartments as a result of periodic boundary conditions. The transmembrane voltage across the membrane can thus be controlled by setting a charge imbalance (dQ in Supplementary Tables 1–4, 7, 8) between the compartments. Due to the low capacity of the membrane, relatively few ions (<5% of the total number) are needed to establish transmembrane voltages of several hundreds of millivolts, so there is only a minor difference in the ion concentration between the compartments[57]. Insertion of one copy of a protein into each bilayer thus enabled the action of both positive and negative voltages to be measured on the protein in a single CompEL system. In practice, each CompEL system was created by stacking two copies of a single-bilayer system taken after certain simulation time (starting time in Supplementary Tables 1–4, 7, 8). To maintain a constant average transmembrane voltage, the algorithm swaps ions between the compartments once an ion permeation event occurs. The intercompartment charge imbalance was checked every 10 ps and averaged over 100 ps; ions were swapped only when the average value deviated by more than one elementary charge from the reference dQ. Systems in which the salt concentration increased to 1 M were created either from the final configuration of a single-bilayer system followed by equilibration or from a single-bilayer system that was originally simulated at the high salt concentration. Full information on the systems, including simulation times and voltages, can be found in Supplementary Tables 1–4, 7, 8.

**Analysis**. Prior to analysis, trajectories were translationally and rotationally fitted onto the transmembrane region of the original protein structure and sampled every 100 ps. To calculate the properties of ions and water in the subunit cavity the region of interest was limited to a box of size 25 Å × 40 Å × 80 Å built around the reference atom of the pore center (Cα of S382 in nhTMEM16 and Cα of C412 in TMEM16K). The box was symmetrical with respect to the reference atom in the y

and z dimensions, but was shifted by 7.5 Å toward the protein periphery in the x dimension to exclude the dimer interface from the analysis. The box was large enough to fully accommodate the interior of the subunit cavity as well as the entrances to it.

Mean conductance was calculated as $G_m = \frac{N_p \cdot e}{V \cdot (t - t_1)}$, where $N_p$ is the total number of permeation events, $e$ is the elementary charge, $V$ is the transmembrane voltage, $t$ is the total simulation time, and $t_1$ is the time for the first permeation event. The part of the simulation prior to the first permeation event was skipped to exclude from the analysis the phase of the initial transition of the pore to the conductive state. Instantaneous conductance was defined as $G_i = \frac{e}{V \cdot \Delta t}$, where $e$ is the elementary charge, $V$ is the transmembrane voltage, and $\Delta t$ is the time between two consecutive permeation events. Defined in this way, the instantaneous conductance characterizes a particular microstate of the pore. Therefore, the median value of the instantaneous conductance, calculated from a protomer trajectory, reports on the microstate that contributes most to the ion conduction. Compared to the standard mean conductance, the median instantaneous conductance is less dependent on the microscopic open probability, and is thus expected to provide the optimal estimate for comparison with experimental single-channel conductances. Therefore, we used median values of the instantaneous conductance to compare our results with existing experimental data.

Local probability density distributions of the lipid phosphorus and nitrogen atoms were calculated based on the position of the corresponding atoms along the outward membrane normal (z dimension of the system). Locality was ensured by considering a limited number of atoms closest to the pore center (four in nhTMEM16 and five in TMEM16K). Hydration of the extracellular part of the pore was calculated as the number of water molecules within a 10 Å region starting from the pore center. In a few cases, extreme dilation of the nhTMEM16 pore in protomers subjected to very high voltages was observed. The dilation was accompanied by a large increase in pore hydration and resulted in a leaky pore with lipid headgroups completely detached from the subunit cavity, similar to the pores previously observed in electroporation simulations[58]. As this behavior was only observed at very high voltages, dilated pores with more than 45 water molecules in the neck region were considered to represent an artificial state of the proteolipidic pore and excluded from the analysis. All distributions were obtained by averaging all (except for dilated) protomers, with first 100 ns of trajectories excluded from the analysis.

The RMSD of the pore region was calculated using the Cα atoms of residues 293–317, 327–348, 374–390, 431–453, 501–518, and 524–543 in nhTMEM16 and of residues 320–349, 355–377, 404–420, 426–449, 496–511, and 518–538 in TMEM16K.

Radii of the first hydration shell of 3.8 Å for $Cl^-$ and 3 Å for $Na^+$ were inferred from the radial distribution functions of the water oxygen atoms surrounding the ions.

The $Na^+$-to-$Cl^-$ permeability ratio was defined as $P_{Na}/P_{Cl} = \frac{G_{Na}}{G_{Cl}}$, where $G$ is the number of permeations made by ions of a certain type by the end of a trajectory. Selectivity classes were defined as follows: $Na^+$ selective, $P_{Na}/P_{Cl} \geq 10$; moderately $Na^+$ selective, $P_{Na}/P_{Cl} \in [2, 10)$; nonselective, $P_{Na}/P_{Cl} \in (0.5, 2)$; moderately $Cl^-$ selective, $P_{Na}/P_{Cl} \in (0.1, 0.5]$, and $Cl^-$ selective, $P_{Na}/P_{Cl} \leq 0.1$.

Sequence alignment was done in Jalview[59] using the ClustalW program[60] on nhTMEM16, afTMEM16, and all murine and human members of the TMEM16 family, excluding TMEM16H.

Lipid localization sites used for analysis of the contacts between the lipid headgroups and a protein were defined based on the distributions of POPC phosphorus and nitrogen atoms and covered the following regions along the pore axis: pc [-3:7], nc [-5:0], pe [7:17], and ne [5:10] in nhTMEM16 and p1 [-15:-10], p2 [-8:-2], p3 [0:5], p4 [10:15], n0 [-18:-13], n1 [-12:-7], n2 [-4:1], n3 [4:9], n4 [8:13], and n5 [13:18] in TMEM16K. Ion-localization sites used for the analysis of contacts between ions and a protein or lipid headgroups were defined based on the distributions of permeating ions and covered the following regions along the pore axis: $Cl_v$ [-16:-1], $Cl_e$ [2:17], $Na_i$ [-25:-15], $Na_v$ [-8:2], and $Na_e$ [5:15] in nhTMEM16 and $Na_i$ [-20:-10], $Na_c$ [-10:0], and $Na_e$ [4:12] in TMEM16K.

Blockage of ion permeation by lipid headgroups was analyzed as follows. When a permeating ion was present in a region corresponding to a specific lipid localization site, its coordinates relative to the reference atom of the pore center were saved. Then the x and y components of the coordinates were averaged over all the visits in all simulations. The distance of a lipid headgroup moiety to the ion permeation pathway in the xy plane was thus defined as the distance from this average ion position. When an ion visited the part either 2.5 Å above (influx) or 2.5 Å below (efflux) the center of the region, the distance from a specific moiety to the permeation pathway was measured if the moiety was present in the region and average distances over the time of the visit were then calculated. Thus, each continuous ion visit to a region that was accompanied by the presence of a headgroup moiety in the same region resulted in one distance value. These distances were then averaged over the whole trajectory to provide a single data point for each independent protomer. This analysis was done separately for permeating and blocked ions to enable an assessment of interference between lipid headgroups and ion permeations.

GROMACS tools and bespoke python scripts using the MDAnalysis library[61] were used for all analyses. Three-dimensional density maps were obtained with GROmaρs[62]. All visualizations were done using PyMOL version 1.8.

**Reporting summary**. Further information on research design is available in the Nature Research Reporting Summary linked to this article.

## Data availability
The datasets derived from MD simulations and underlying the figures are available in a Source Data file. Raw data are archived at Jülich Supercomputing Centre and RWTH Aachen University and are available from the corresponding author upon reasonable request.

## Code availability
GROMACS tools and bespoke python scripts using the MDAnalysis library were used to analyze molecular dynamics trajectories. The python script used to analyze ion conduction is freely available at https://jugit.fz-juelich.de/computational-neurophysiology/scramblase_ion_conduction

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

## Acknowledgements

We thank Diana Kondinskaia and Drs. Christoph Fahlke, Claudia Alleva, and Yulia Kolobkova for helpful discussion and critically reading the manuscript. This work was funded by the Deutsche Forschungsgemeinschaft (German Research Foundation) to J.-P. M. (MA 7525/1-2, as part of the Research Unit FOR 2518/DynIon, project P4; and MA 7525/2-1, as part of the Research Unit FOR 5046, project P2), the Jülich-Aachen Research Alliance Center for Simulation and Data Science (JARA-CSD) School for Simulation and Data Science (SSD), and by a grant from the Interdisciplinary Centre for Clinical Research within the faculty of Medicine at the RWTH Aachen University (IZKF TN1-3/IA532003). The authors gratefully acknowledge the computing time granted through JARA on the supercomputer JURECA at Forschungszentrum Jülich.

## Author contributions

J.-P.M. conceived and supervised the project. J.-P.M. and A.K. designed the research. A.K. produced and analyzed data, and drafted the manuscript. A.K. and J.-P.M. wrote the final version of the paper.

## Funding

## Competing interests

The authors declare no competing interests.
