## [Peer Review File · Nature Communications]

Reviewer #1 (Remarks to the Author):

The paper presents a detailed and comprehensive investigation of mechanisms involved into ion permeation and selectivity in TMEM16 lipid scramblases. The authors used computational electrophysiology approach and determined the conducting state of TMEM16. They further studied the role of lipids on ion permeation and selectivity by performing the simulations with different lipid compositions. They simulated different forms of TMEM16 to test whether the proposed mechanism is generally applicable. Overall, the paper is well and convincingly written and the analysis was performed properly. Below I will summarize list of my suggestions that may require additional attention before re-submission of this paper:

1. In this work, the authors calculated the conductance and instantaneous conductance of TMEM16. While the conductance can be considered as a standard parameter for studying ion channels, instantaneous conductance is rather a new parameter proposed by the authors. Therefore, I would suggest the authors to include the "normal" conductance in the Figure 1 as well. Furthermore, I hope the authors can provide a rationale to the usage of instantaneous conductance, rather than only stating that the instantaneous conductance fits better to the single-channel data. This will be important for other researchers in the field to use this parameter in their future study.
2. I suggest the authors to include the length of total simulations time, the choice of force field and the applied voltages in mV in the main text. The authors mentioned in the Materials and Methods section that the part of the simulation prior to the first permeation event was skipped to exclude from the analysis. Can the author provide the time length of this initial phase?
3. The authors determined a few cation and anion binding sites along the pore axis. Can the authors provide the dwell time of the ion binding from the simulations? Is the dwell time very different for cation and anion?
4. Is it known experimentally that only sodium permeates through the channel? Why do the authors not simulate potassium or a mixture of potassium and sodium?

Reviewer #2 (Remarks to the Author):

In this manuscript, the authors conducted atomistic molecular dynamics (MD) simulations to study the mechanism ion conduction through the dual functional TMEM16 lipid scramblases/ion channels. How a single permeation pore (proteolipidic) simultaneously permeates two distinct substrates, ions and phospholipids, has been an intriguing question in the field. The recent structural, functional and computation studies have advanced the understanding the TMEM16 proteins, especially how phospholipids go through the pore. This manuscript attempted to use computational methods to understand how ions go through the pore when the pore is also occupied by the phospholipid headgroups. The computational findings are interesting and provides some insights into the mechanism of dual ion/lipid permeation in TMEM16 proteins. However, there are serious issues in the manuscript, which dampen the enthusiasm of this reviewer.

Major points.

1. This is a pure computational study without any functional support. Although the authors

attempted to cite some functional publications to support their findings, the authors seem not fully understand those publications or only cherry-pick part of the findings from these functional studies that fit their conclusions. For instance, 1) line 129-132, the authors tried to cite a recent paper (ref 38) to support their simulation findings on voltage dependent reorientations of the lipid headgroups in the pore. The cited paper used one single nanosecond electric pulsed (300 ns, 25.5KV/cm) to activate TMEM16F scramblase. However, the high voltage pulse was used to induce membrane nanoporation, which caused Ca influx and then activated TMEM16F. The 300 ns pulse (shorter than the simulations in this paper) was not used to activate TMEM16F. Indeed, if no Ca is present, the voltage pulse cannot activate TMEM16F scramblase. 2) In line 92-100, when the authors tried to make sense their calculated single channel conductance for nhTMEM16, they compared with that of TMEM16F and afTMEM. Despite the fact that the median values of 31.3 pS for nhTMEM16 (Fig. 1e), the author only compared the conductance of ~100 pS in rare events with afTMEM's 300 pS conductance and claimed that they have "the same order of magnitude". In addition, the authors specifically ignored the report that TMEM16F has tiny single channel conductance (< 1pS, Yang, H. et al, Cell, 2012). Instead, they only cited a paper reported TMEM16F has 50 pS conductance, which was based on low quality single channel recording (Martins, J. R, PNAS, 2011) and never got repeated ever since. 3) Line 254-256, when the authors tried to fit their findings on salt concentration-dependent change of ion selectivity with experimental data of TMEM16F (Ye, et al, eLife, 2019), they forgot a fundamental difference between their computational condition with the experimental conditions. Ye et al used 15, 45 and 150 mM NaCl to measure the dynamic change of ion selectivity using patch clamp, whereas the authors used NaCl change from 250 mM to 1000 mM. With such high concentration of salts, it won't be surprising to see surface charge/screening effect in simulation. If the authors really want to make comparison, please use low salt concentration instead.

2. In addition, plenty of mutagenesis studies have identified key residues that disrupt lipid scrambling and ion permeation. It is disappointing that the authors did not use computational methods to examine any single one of them to support their conclusions.

3. Considering the senior author is also an established experimentalist, it is reasonable to expect some functional evidence to support their conclusions. Otherwise, the manuscript might be more appropriate to a computation-oriented journal.

4. Structural templates used were not well justified. Although different structural conformations have been solved for the fungal nhTMEM16 and mammalian TMEM16K, their ion channel activities have not been systematically characterized (selectivity and conductance are unknown). This makes it very difficult to compare the authors' computational estimations about ion conduction and selectivity with experimental data. TMEM16F has been thoroughly characterized using patch clamping. However, the apo and Ca bound TMEM16F structures are not fully open. Nevertheless, another fungal TMEM16 (afTMEM16) has recently solved in different conformational states (Falzone, M., et al, eLife, 2019). As the authors mentioned in the manuscript, afTMEM16 is known to have 300 pS and PK/PCI of 1.5 (Malvezzi, M., 2013, Nat. Commun.). This TMEM16 would be perfect for the authors to compare their computational findings with the reported functional finding. However, it was not justified why afTMEM16 was not chosen.

5. Experimental details were not well disclosed. It has been extremely difficult for the reviewer to figure out the exact simulation conditions for almost all figures. Starting from Figure 2, neither the legend nor the text mentioned which structures were used for simulations, which phospholipids

were in the system, what concentration of NaCl was used, what voltage was imposed to the membrane. Without these information, it's difficult to evaluate the results.

6. To this reviewer, the authors treated the phospholipid headgroups as static components of the pore. Despite the differences between the headgroups vs full length phospholipids and the differences between artificial lipid environment vs real cell membrane, a functional TMEM16 scramblase would constantly transport phospholipids with high speed, ie the headgroups are constantly and dynamically moving. It is hard to imagine how Na and Cl ion would have fixed contact with the headgroups and stay at preferred locations in the pore. The authors had a perfect chance to investigate this dynamic dual transport when simulating with PC/PS and PE/PG. Unfortunately, none of these simulations was clearly described. In addition, it is unknown how ion conduction change when PS or PG is present. It is expected that the negatively charged headgroups from these lipids will be dramatically influenced by changing voltage. They would also exert larger effects on governing ion selectivity than the neutral PC headgroups. Is this the case?

7. The authors mentioned multiple times that in their simulations, the anionic lipids were excluded from the central part of pore. What does this mean? PS and PG headgroups transient hop through the center of the pore or they have a different permeation path from PC (out of groove)?

8. Does the ion conductive intermediate conformation also permeate phospholipids? If phospholipids are excluded from the pore in this state, it means the scramblases can form a protein-only ion conductive pores, which is contradictory to the proteolipidic pore model proposed in the study.

9. How membrane's electrical field affects the orientation of the permeation pathway's lining residues and lipid headgroups is intriguing. However, it would be more informative if the authors elucidate whether the lipid headgroups (PC, PS and PG) prefer upward movement in concurrence with their upward orientation in positive voltage (and vice versa in the negative voltage). The authors listed almost all charged residues in Fig. 7B, but missed out several conserved charged residues in Fig. 7C (D503, E506 for nhTMEM16 and D497 and E500 for TMEM16K). Why? Are they important for ion conduction? Which directions do their sidechains point to?

10. The authors emphasized that lipid and voltage can dictate the ion permeability and selectivity, what about the role of voltage in determining lipid selectivity, permeability and direction of movement?

11. The description about E313/E318/R432 interacting with lipid headgroups require a revision. Lee et al., Nat Comm 2018, which examined and described the roles of this triad in coordinating phospholipids, utilized POPE and POPG instead of POPC. Difference in lipid headgroups may affect the their interactions with the residues lining the permeation pathway.

12. This study underscores the importance of proteolipidic pore and lipid lining in determining ion permeability and selectivity. However, how would the author explain the ion conductive nhTMEM16 mutations, such as L302A (Khelashvili et al., Nat Comm 2019), that do not scramble lipids? How ion permeability and selectivity are defined in those non-scrambling cases? The non-scrambling mutations demonstrate that the widely open and proteolipidic pore might not be the only requirement for ion permeation in the scramblases as described by the authors.

13. In general, the authors need to tune down their conclusions as neither their own findings nor previously published results fully support their conclusions.

There are many minor issues in the manuscript. The reviewer only lists a couple here.

1. In Figure 5 and Figure 7, the authors used the numbers of protomers that display different PNa/PCl to compare their ion selectivity. While this is clear, it would be better to show the PNa/PCl values. Would the weighted PNa/PCl based on the number of protomers be calculated and presented?

2. When the authors described their results in text, often times they only mentioned figure number without pointing to the exact panels. For instance, when referencing the supplementary figure7, it would help the readers to refer to specific panels.

Reviewer #3 (Remarks to the Author):

The manuscript describes the results of a very detailed investigation of ion permeability in TMEM16 scramblases (nhTMEM16, TMEM16K and TMEM16F) from the application of a method considered to represent “computational electrophysiology simulations”, termed CompEL. The findings support the proteolipidic-pore mechanism with a fully open groove as a main permeation route, against the alternating pore-cavity model. Detailed reports of the apparent role of lipid headgroups in shaping this ion pathway and controlling the permeability are likely to advance thinking in the field. Ion selectivity changes are shown to change with lipid composition or salt concentration and for TMEM16K, residue types in the groove and protein structure suggest that TMEM16K is a cation-selective ion channel.

The conclusions are sufficiently intriguing to engender interest, which should be sufficient to eliminate the for presenting them in a manner suggesting that the aim of the manuscript is to refute the connection between ion permeation and the intermediate state. In reality the prevalent view in the field based on specific evidence is that the intermediate state does not transport lipids but still can transport ions. The results presented in the manuscript are sufficiently interesting and do not require the establishment of a false dichotomy, so that a reconsideration of statements such as .. is recommended.

Some concerns to be addressed:

A. The manuscript is based on the use of the CompEL molecular dynamics simulation method. This is not a widely known or used approach and thus requires much more definition and discussion of assumptions and considerations than is afforded in the manuscript. The very setup of the simulated system for which all the results are reported is provided only in the Supplementary Materials (Fig. 1a there). This alone raises two important issues that merit discussion and clarification in the manuscript

A1. The title and text refers to computational electrophysiology. This may, or may not be an apt description of the method but a brief review of why and to what extent this approximation is considered to reproduce insights from electrophysiology experiments for this type of system (which

may not be a gated channel of the kind discussed previously by this group) is in order, because the language describing the experiments is taken from actual electrophysiological measurements, which could be very misleading (e.g., line 82 “induced by the applied voltage”) when the simulation is carried out in an ion gradient established in a box).

A2. The setup description in Supplementary Fig. 1a shows that the EC and IC ends of the two proteins sampled are embedded in water compartments in which the cation is the same (both in the middle and at the extreme ends of the box). That, and the asymmetry of the membrane would be different in an electrophysiology experiment. While these may be acceptable approximations, they need to be presented as such. In particular, would this symmetry be expected to affect the influx/efflux results for the ions in Figure 4?

B. The manuscript refers to structures and mechanistic studies of nhTMEM16 in which intermediate and open groove states were described in detail. But the widening of the structure in the presented simulations is described by RMSD measures. Could this be shown in more details? For example, by calculating changes in pore radius and/or by following in the trajectories the distances between landmark residues Y432-T333, and between the E313, E318, and R432 triad residues that emerged as mechanistically essential. Indeed, it would be good to present structural details from the simulations in the manuscript regarding the following

(i)- How the pair-wise distances between the residues in the E313/E318/R432 triad change in the simulations

(ii)- What conformations these residues adopt when ions bind to the EC side and get transported

(iii)-How the “widened” structure compares to the L302A structure of this TMEM16, and whether it conduct lipids

(iv)-Whether lipid scrambling events were observed in the simulations and if so, what the relationship might be between the scrambling events and ion permeation events.

C. A key point made in the manuscript concerns lipid head group orientations in negative/positive voltages. While it is clear that the lipid headgroups would change directions in the different potential fields it would be informative for a description of a scramblase mechanism to connect these results to the dynamics of lipid movement to clarify the interpretation. Incidentally, it appears that the error bars in Figure 2d are large and the data for zero and negative potentials basically overlaps. How can this be interpreted?

D. Was the analysis for Figure 2 done only for the fully open state (starting from 4WIS). It seems necessary to do it as well for the intermediate state (starting 6QMA), because conductive protomers were identified there as well (Figure 1) and information is needed regarding the lining of the pore with lipid headgroups in order to generalize about the requirement for a proteolipidic pore to observe ion permeation, as done in the manuscript. The same reasoning would suggest the need for analysis of the intermediate state to reach the conclusions regarding lipid realignment from Figure 3.

E. The intent and logic of the following phrase in the Discussion seems unclear to this reader. Is the idea that fully activated TMEM16 are some sort of “super-channels”?

“Given that in our simulations single-channel conductance

374 of nhTMEM16 in the intermediate state (~10 pS) was comparable to that of TMEM16A

375 (~3 - 8 pS), it is tempting to speculate that ion conduction in the intermediate state

376 resembles that of bona fide channels of the family, whereas a fully open subunit cavity is

377 necessary for complete activation of ion conduction in TMEM16 lipid scramblases.”

F. An idea presented in the Discussion as a hypothesis appears to be a guess or speculation not really based on anything specific: “We hypothesize
427 that the deranged Ca²⁺ signaling leading to spinocerebellar ataxia in patients with
428 mutations in TMEM16K41 could, at least partly, result from impaired function of TMEM16K
429 as ER-localized cation-selective ion channel..
This is followed by a further speculation about the consequences. Unless labeled as a speculation, some clarification for the basis for this hypothesis would seem to be necessary here.

Response to the reviewers' comments

We thank all the reviewers for providing valuable comments and questions that helped us to improve the manuscript. Changes in the manuscript text are highlighted in blue and figure panels with newly added data are marked by colored description in figure captions.

Reviewer #1 (Remarks to the Author):

The paper presents a detailed and comprehensive investigation of mechanisms involved into ion permeation and selectivity in TMEM16 lipid scramblases. The authors used computational electrophysiology approach and determined the conducting state of TMEM16. They further studied the role of lipids on ion permeation and selectivity by performing the simulations with different lipid compositions. They simulated different forms of TMEM16 to test whether the proposed mechanism is generally applicable. Overall, the paper is well and convincingly written and the analysis was performed properly. Below I will summarize list of my suggestions that may require additional attention before re-submission of this paper:

We thank the reviewer for the positive evaluation of our work and for providing valuable suggestions.

1. In this work, the authors calculated the conductance and instantaneous conductance of TMEM16. While the conductance can be considered as a standard parameter for studying ion channels, instantaneous conductance is rather a new parameter proposed by the authors. Therefore, I would suggest the authors to include the "normal" conductance in the Figure 1 as well. Furthermore, I hope the authors can provide a rationale to the usage of instantaneous conductance, rather than only stating that the instantaneous conductance fits better to the single-channel data. This will be important for other researchers in the field to use this parameter in their future study.

Due to high temporal resolution of the simulations, we can detect pore microstates with different ion-conductance levels. For instance, when a central lipid headgroup blocks the ion-permeation pathway, the nhTMEM16 pore is temporarily closed. Since the standard mean conductance is defined as number of permeation events divided by the simulation time, it reflects an average over all such microstates including the closed states. By contrast, the instantaneous conductance (computed from the waiting times between successive permeation events) permits direct characterization of the microstate that the pore adopts during ion-permeation events. We note that the analysis of simulated ion currents using inverse waiting times has been established before (Jensen, M.Ø., et al, PNAS, 2010, 107, 5833-5838; Jensen, M.Ø., et al, J Gen Physiol, 2013, 141, 619-632). Thus, the median value of the instantaneous ion conductance in a simulation trajectory reports on the dominant microstate which contributes most to ion conduction.

Importantly, whereas estimates of the mean conductance critically rely on correct sampling of the pore-microstate distribution, the median instantaneous conductance is expected to be less dependent on conformational sampling in our simulations. Moreover, microstates, with instantaneous conductances much lower than the median, can be assigned to the nonconductive state of the pore (related to the current discretization in the analysis of experimental single-channel recordings); whereas such microstates contribute to the standard mean conductance in our simulations, they are more related to open probability than to the single-channel conductances in the analysis of experimental data. Therefore, we believe that—as opposed to the standard mean conductance—the median instantaneous conductance is the most reliable and optimal estimate of the experimental single-channel conductance from our simulations. Nevertheless, our conclusion that the fully open state represents the main ion-conductive state does not depend on the analysis method, with both standard and instantaneous ion conductances of nhTMEM16 being lower in the intermediate than in the fully open state. This is now discussed on lines 643–651 in the Methods section, and the distribution of the mean conductance over the simulation replicas is now included as an inset

in Figure 1d.

2. I suggest the authors to include the length of total simulations time, the choice of force field and the applied voltages in mV in the main text. The authors mentioned in the Materials and Methods section that the part of the simulation prior to the first permeation event was skipped to exclude from the analysis. Can the author provide the time length of this initial phase?

We have added information on simulation time, force field, and applied voltages to the Results section (lines 79, 94–95, 334–336).

The latencies until the first permeation event for each conductive protomer are now shown in Supplementary Fig. 3d and Supplementary Fig. 4a for wild-type and L302A nhTMEM16, respectively. In the wild type, the median latencies were ~ 209 ns and ~ 77 ns in the intermediate and fully open conformations, respectively. These data are consistent with our observation that the pores in the intermediate state need to undergo more prominent (and therefore slower) conformational changes than those in the fully open state to permit ion conduction, consistent with our conclusion that the fully open conformation represents the main ion-conductive state. This is now discussed on lines 138–140.

3. The authors determined a few cation and anion binding sites along the pore axis. Can the authors provide the dwell time of the ion binding from the simulations? Is the dwell time very different for cation and anion?

We have calculated the maximum dwell time of ions at their accumulation sites in each protomer trajectory for different lipid membranes; these data are now shown in Supplementary Fig. 9a and Supplementary Fig. 15c,d. In POPC membranes, the median values calculated over all the nhTMEM16 protomers range from 4.3–8.6 ns, with only minor differences between cations and anions (Supplementary Fig. 9a). Notably, in the protomers embedded in either POPC:POPS or POPE:POPG membranes, the maximum dwell time of Na^+ ions at the extracellular site was noticeably increased (Supplementary Fig. 15c,d), suggesting that anionic lipids stabilize Na^+ ions at this site, consistent with the observed increase in Na^+ -to- Cl^- selectivity in these simulations. These data are now discussed on lines 217–220 and 308–309.

4. Is it known experimentally that only sodium permeates through the channel? Why do the authors not simulate potassium or a mixture of potassium and sodium?

According to the existing experimental data nhTMEM16 is permeable to both Na^+ and K^+ ions (Jiang et al, Elife, 2017; Lee et al, Biophysical Journal, 2016). We decided to focus on the anion/cation selectivity rather than on relative cation selectivities, the former of which was expected to be more sensitive to membrane lipid composition; therefore, we used only Na^+ as cation to obtain best possible statistics of Na^+ and Cl^- conduction. Finally, Na^+ and K^+ ions differ mainly in size and hydration energy, but we demonstrated that both small Na^+ and larger Cl^- ions are conducted in the fully hydrated states (Supplementary Fig. 12a for nhTMEM16; Supplementary Fig. 18b for TMEM16K). Thus, we expect that the general conduction mechanisms are valid for K^+ ions as well.

Reviewer #2 (Remarks to the Author):

In this manuscript, the authors conducted atomistic molecular dynamics (MD) simulations to study the mechanism ion conduction through the dual functional TMEM16 lipid scramblases/ion channels. How a single permeation pore (proteolipidic) simultaneously permeates two distinct substrates, ions and phospholipids, has been an intriguing question in the field. The recent structural, functional and computation studies have advanced the understanding the TMEM16 proteins, especially how phospholipids go through the pore. This manuscript attempted to use computational methods to understand how ions go through the pore when the pore is also occupied by the phospho-

lipid headgroups. The computational findings are interesting and provides some insights into the mechanism of dual ion/lipid permeation in TMEM16 proteins. However, there are serious issues in the manuscript, which dampen the enthusiasm of this reviewer.

We thank the reviewer for finding our results interesting and for providing thoughtful comments, which we have fully addressed below.

Major points.

1. This is a pure computational study without any functional support. Although the authors attempted to cite some functional publications to support their findings, the authors seem not fully understand those publications or only cherry-pick part of the findings from these functional studies that fit their conclusions.

Extensive multireplica simulations under near-physiological conditions together with the Computational Electrophysiology method allowed us to quantify typical functional properties of ion channels such as ion conductance and selectivity. Importantly, we directly connected these functional characteristics of TMEM16 lipid scramblases with their molecular structures and provided molecular-level mechanistic explanations for many experimental observations (including the prominent variability of single-channel conductances reported in different studies and salt concentration-dependent shifts in ion selectivity in TMEM16 lipid scramblases). Of note, in most of the experimental studies, many microscopic parameters of the studied system remain unknown, limiting direct quantitative comparison between experimental and computational results. Nevertheless, we designed our simulations trying to approach experimental conditions as closely as possible, such that our results can be compared with published functional data, and our results indeed agree very well in a semi-quantitative way with the existing experimental data on the functional properties of TMEM16 lipid scramblases.

For instance, 1) line 129-132, the authors tried to cite a recent paper (ref 38) to support their simulation findings on voltage dependent reorientations of the lipid headgroups in the pore. The cited paper used one single nanosecond electric pulsed (300 ns, 25.5KV/cm) to activate TMEM16F scramblase. However, the high voltage pulse was used to induce membrane nanoporation, which caused Ca influx and then activated TMEM16F. The 300 ns pulse (shorter than the simulations in this paper) was not used to activate TMEM16F. Indeed, if no Ca is present, the voltage pulse cannot activate TMEM16F scramblase.

We agree that the connection that we implied could be misleading, since the details of the voltage effects on lipid scrambling in fully activated Ca^{2+} -bound TMEM16F await further experimental investigation. We have now removed this citation.

2) In line 92-100, when the authors tried to make sense their calculated single channel conductance for nhTMEM16, they compared with that of TMEM16F and afTMEM. Despite the fact that the median values of 31.3 pS for nhTMEM16 (Fig. 1e), the author only compared the conductance of 100 pS in rare events with afTMEM's 300 pS conductance and claimed that they have "the same order of magnitude". In addition, the authors specifically ignored the report that TMEM16F has tiny single channel conductance ($< 1\text{pS}$, Yang, H. et al, Cell, 2012). Instead, they only cited a paper reported TMEM16F has 50 pS conductance, which was based on low quality single channel recording (Martins, J. R, PNAS, 2011) and never got repeated ever since.

We thank the reviewer for pointing this out. Indeed, in our simulations the median value of ion conductance demonstrated by nhTMEM16 in the fully open state is lower than the single-channel conductance of afTMEM16 determined experimentally. However, our comparison of a few protomers with ion conductance of $> 100\text{ pS}$ with afTMEM16 was to stress that nhTMEM16 in the fully open but not in the intermediate state could reach ion conductances comparable to that

of afTMEM16. We have now clarified this on lines 126–126.

Similarly, we compared our results with experimental data on TMEM16F to demonstrate that the cited recordings (Martins et al., PNAS 2011) would likely correspond to the fully open state of the scramblase, assuming that Ca^{2+} -bound TMEM16F adopts states similar to nhTMEM16 (i.e. closed, intermediate, and fully open). We thank the reviewer for reminding us of the study by Yang et al. (Cell, 2012), where TMEM16F was reported to have rather small ion conductance, comparable to that of bona fide TMEM16 channels and to nhTMEM16 in the intermediate state. We, therefore, suggest that these recordings by Yang et al. likely correspond to the intermediate state of TMEM16F, and that the equilibrium between different ion-conductive states of TMEM16 lipid scramblases might be subject to the experimental conditions, as now discussed on lines 119–123.

3) Line 254-256, when the authors tried to fit their findings on salt concentration-dependent change of ion selectivity with experimental data of TMEM16F (Ye, et al, eLife, 2019), they forgot a fundamental difference between their computational condition with the experimental conditions. Ye et al used 15, 45 and 150 mM NaCl to measure the dynamic change of ion selectivity using patch clamp, whereas the authors used NaCl change from 250 mM to 1000 mM. With such high concentration of salts, it won't be surprising to see surface charge/screening effect in simulation. If the authors really want to make comparison, please use low salt concentration instead.

In addition to the precise salt concentration, the computational and experimental conditions differ in the local membrane composition, which can strongly affect the observed ion selectivity, as we have shown in our study. Since the local membrane composition is usually unknown in patch-clamp experiments, we believe that these experimental results can be compared with our simulation results only in a semi-quantitative manner. However, considering that the reduction of PNa/PCl upon increases in mono-, di-, or trivalent salt concentrations is the main observation by Ye et al., our computational results do agree very well with these experiments. Moreover, in our simulations we were able to finely control the membrane lipid composition, so that we could identify lipids as the main determinant for the observed ion-selectivity shift.

Since monovalent cations have a lower affinity to lipid membranes compared to divalent cations, we originally used 1000 mM NaCl solution to mimic the effect of Ca^{2+} on ion selectivity, as observed by Ye et al. Nevertheless, to further approach the experimental conditions with our simulations, we have now conducted a series of simulations with a reduced bulk NaCl concentration of 200 mM and added Ca^{2+} ions to reach a bulk concentration of ~ 1 mM (similar to the experimental conditions in Ye et al.). Similar to the effect of increasing the NaCl concentration (Fig. 5), addition of Ca^{2+} lead to a prominent reduction in PNa/PCl (Supplementary Fig. 16, Supplementary Table 8).

We also agree with the reviewer that there is no surprise in the screening of the membrane charge by 1000 mM NaCl. Indeed, we used such high concentration to prove that the screening could lead to a shift in ion selectivity towards the values observed in a neutral membrane. However, it was not the screening itself, but rather the lipid effects on the observed ion selectivity that we found intriguing. This result is now discussed in more detail on lines 320–323.

2. In addition, plenty of mutagenesis studies have identified key residues that disrupt lipid scrambling and ion permeation. It is disappointing that the authors did not use computational methods to examine any single one of them to support their conclusions.

We have now conducted a series of simulations of L302A nhTMEM16. This mutant was structurally resolved in the intermediate state using cryo-EM and was shown to have strongly impaired scrambling activity but only slightly reduced ion-channel activity in liposome experiments (Khelashvili et al, Nat. Commun., 2019). In our simulations, the L302A mutant demonstrated ion conductance and overall conduction probability lower than those of the wild type protein in the fully open state. Similar to the wild-type in the intermediate state, the mutant had only a negligible fraction of lipid headgroups at the central part of the ion-conducting pore (Supplementary Fig. 6a,b), supporting our conclusion that the proteolipidic pore represents the main ion-conductive

state of the nhTMEM16 pore, characterized by the highest ion-conduction competence. This is now discussed on lines 110–114, 161–164, and 437–441.

3. Considering the senior author is also an established experimentalist, it is reasonable to expect some functional evidence to support their conclusions. Otherwise, the manuscript might be more appropriate to a computation-oriented journal.

In our study, we identified the molecular mechanisms of ion conduction and ion selectivity in TMEM16 lipid scramblases. Not only do our results agree very well with the existing experimental data, they also significantly augment them with mechanistic insights at atomic resolution. Using state-of-the-art simulations we could directly connect these details to the functional properties of TMEM16 lipid scramblases. Furthermore, we described direct effects that lipids can have on functional properties of ion channels. Therefore, we believe that our findings are not only relevant to computational scientists, but primarily to structural biologists, biophysicists, physiologists, and molecular biologists interested in mechanisms of ion channels and lipid scramblases.

4. Structural templates used were not well justified. Although different structural conformations have been solved for the fungal nhTMEM16 and mammalian TMEM16K, their ion channel activities have not been systematically characterized (selectivity and conductance are unknown). This makes it very difficult to compare the authors' computational estimations about ion conduction and selectivity with experimental data. TMEM16F has been thoroughly characterized using patch clamping. However, the apo and Ca bound TMEM16F structures are not fully open. Nevertheless, another fungal TMEM16 (afTMEM16) has recently solved in different conformational states (Falzone, M., et al, eLife, 2019). As the authors mentioned in the manuscript, afTMEM16 is known to have 300 pS and PK/PCL of 1.5 (Malvezzi, M., 2013, Nat. Commun.). This TMEM16 would be perfect for the authors to compare their computational findings with the reported functional finding. However, it was not justified why afTMEM16 was not chosen.

In our study we simulated all the mammalian TMEM16 lipid scramblases, whose structures are currently resolved (TMEM16F and TMEM16K). We showed that the Ca^{2+} -bound structure of TMEM16F represents a nonconductive state of the protein and that the fully open cavity of human TMEM16K conducts ions through a well-structured proteolipidic pore. As correctly noted by the reviewer, a detailed experimental characterization of the ion-conduction properties of TMEM16K is still to come, but we believe that our results would additionally foster such investigation.

Concerning the fungal homologs of the family, our choice of nhTMEM16 over afTMEM16 was based on several reasons. First, three different conformational states (closed, intermediate and fully open) of Ca^{2+} -bound nhTMEM16 have been structurally characterized (Kalienkova et al., eLife 2019). In contrast, only the fully open conformation of Ca^{2+} -bound afTMEM16 was resolved by cryo-EM (Falzone et al., eLife 2019). The availability of structures for different functional states of nhTMEM16 was instrumental for us to establish the ion conduction mechanism in TMEM16 lipid scramblases and to demonstrate that the fully open conformation represents the main ion-conductive state. Second, the structure of nhTMEM16 in the fully open state (PDB ID: 4WIS; resolution of 3.30 Å) is more complete and has a higher resolution than the structure of afTMEM16 in the fully open state (PDB ID: 6E0H; resolution of 4.05 Å). Third, nhTMEM16 has been extensively studied both computationally and experimentally (Lee, B.-C., et al, Biophysical Journal, 2016; Bethel, N.P., et al, PNAS, 2016; Jiang, T., et al, eLife, 2017; Lee, B.-C., et al, Nat. Commun., 2018; Khelashvili, G., et al, Nat. Commun., 2019; Kalienkova, V., et al, eLife, 2019), allowing for the comparison with already known functional properties (e.g. scrambling and ion activities of the intermediate and fully open states).

Although we simulated only the fungal homolog nhTMEM16, its structural similarity with afTMEM16 (sequence similarity of 73% for the transmembrane part) permits functional predictions for afTMEM16 based on our results. In particular, among the residues, which we have demonstrated to be involved in nhTMEM16 ion conduction, two are not conserved between nhTMEM16 and

afTMEM16: N435 and D367 in nhTMEM16 are substituted with K428 and K359 in afTMEM16. Additional positive charge brought by these two basic residues would suggest higher preference for anions in afTMEM16 compared with nhTMEM16. Indeed, when reconstituted in an anionic PE:PG membrane, nhTMEM16 has PNa/PCl of 8.7 (our simulations) and afTMEM16 had PK/PCl ~ 1.5 (Malvezzi et al. Nat. Commun. 2013). In a neutral membrane, we expect afTMEM16 to demonstrate PK/PCl < 1 . The ion selectivity of afTMEM16 is now discussed on lines 297–304.

Although single-channel conductance of nhTMEM16 has not been experimentally measured yet, the structure and sequence similarity suggest it to be comparable with that of afTMEM16. Ion conductance of afTMEM16 was measured in a PE:PG membrane (Malvezzi et al. Nat. Commun. 2013), and we have now calculated that of nhTMEM16 in a membrane of the same composition (Supplementary Fig. 13b). The ion-conductance level of nhTMEM16 in the fully open conformation turned out to be similar to that in a neutral POPC membrane; the median value was of ~ 32 pS and a few protomers demonstrated ion conductance of ~ 100 pS (Supplementary Fig. 13b). Although the experimental single-channel conductance of afTMEM16 is higher than the ion conductance of nhTMEM16 in our simulations, these results indicate that the fully open conformation corresponds closer than the intermediate one to the experimentally characterized state of afTMEM16, as discussed on lines 123–126. We now added data on ion conductance of nhTMEM16 in the PE:PG membrane to Supplementary Fig. 13.

5. Experimental details were not well disclosed. It has been extremely difficult for the reviewer to figure out the exact simulation conditions for almost all figures. Starting from Figure 2, neither the legend nor the text mentioned which structures were used for simulations, which phospholipids were in the system, what concentration of NaCl was used, what voltage was imposed to the membrane. Without these information, it's difficult to evaluate the results.

Although all the simulation conditions were fully described in the main text and supplementary tables, we apologize for the confusion. To make it easier to connect the results to particular conditions, we have now additionally specified them on lines 89–91, 94–95, and 334–336.

6. To this reviewer, the authors treated the phospholipid headgroups as static components of the pore. Despite the differences between the headgroups vs full length phospholipids and the differences between artificial lipid environment vs real cell membrane, a functional TMEM16 scramblase would constantly transport phospholipids with high speed, ie the headgroups are constantly and dynamically moving. It is hard to imagine how Na and Cl ion would have fixed contact with the headgroups and stay at preferred locations in the pore. The authors had a perfect chance to investigate this dynamic dual transport when simulating with PC/PS and PE/PG. Unfortunately, none of these simulations was clearly described. In addition, it is unknown how ion conduction change when PS or PG is present. It is expected that the negatively charged headgroups from these lipids will be dramatically influenced by changing voltage. They would also exert larger effects on governing ion selectivity than the neutral PC headgroups. Is this the case?

Although TMEM16 lipid scramblases transport lipids with high rate ($> 2 \cdot 10^4$ lipid s^{-1} ; Malvezzi et al. Nat. Commun. 2013; Lee et al. Nat. Commun. 2018), ion conduction through the proteolipidic pore is a much faster process ($\sim 2 \cdot 10^7$ ion s^{-1} at 100 mV, when assuming a 30-pS ion conductance). Moreover, our analysis of the dwell time of ions and lipid headgroups at their accumulation sites (now shown in Supplementary Fig. 9) demonstrated a clear time-scale separation between ion and lipid transport, as now discussed on lines 217–220. We did demonstrate that the permeating ions have on average notable probability (40–50%) to interact with the pore-lining lipid headgroups (Supplementary Fig. 10a,b and 18c). However, this does not imply fixed contacts between permeating ions and the lipid headgroups, considering the one-order-of-magnitude difference in the dwell times of ions and lipids and their sites (Supplementary Fig. 9).

The clear time-scale separation between lipid scrambling and ion conduction, as well as more than order of magnitude difference in mass of ions and lipid molecules, predicts a rather weak

coupling between the two transport processes and suggest that ions would unlikely engage lipids into concomitant transport. Moreover, we did not find any effect of ion binding to the extracellular sites on the conformation of the E313/E318/R432 triad (see response to the comment B of reviewer 3), which has been demonstrated to play an important role in the lipid-scrambling mechanism (Lee et al, Nat. Commun., 2018). However, in contrast to hundreds of ion-permeation events, only six lipid-scrambling events were observed in our simulations (all in the systems with fully open nhTMEM16 in a POPC membrane), precluding us from definite conclusions on the effect of ions on lipid transport.

Since an anionic lipid lacks dipole moment associated with its headgroup, its orientation in the pore would not be directly affected by the direction of the applied electric field. Nevertheless, the reviewer is right and presence of anionic lipids does change the ion selectivity of the pore, as we showed in this study (Fig. 5). However, in contrast to POPC lipids, which affect selectivity through the orientation of their lipid headgroups at the central part of the pore, anionic lipids alter the ion selectivity by bringing negative charge to the pore entrances.

We have now complemented our description of ion conduction mediated by nhTMEM16 in anionic membranes with the analysis of ion conductance, distribution of permeating ions along the pore, and their dwell times at the localization sites. The results are now shown in Supplementary Fig. 13b and Supplementary Fig. 15; nhTMEM16 ion conduction in anionic membranes is now discussed on lines 304–309.

7. The authors mentioned multiple times that in their simulations, the anionic lipids were excluded from the central part of pore. What does this mean? PS and PG headgroups transient hop through the center of the pore or they have a different permeation path from PC (out of groove)?

Our data indicate that the presence of the anionic lipids in the central part of the cavity is much lower than that of neutral POPC and POPE (Supplementary Fig. 14a,b), suggesting lower scrambling rates of POPS and POPG compared to POPC and POPE in agreement with experimental data (Malvezzi et al, Nat. Commun., 2013). Since we did not observe scrambling of anionic lipids in our simulations, we cannot rule out an out-of-the-groove scrambling mechanism; however, we believe that during scrambling, anionic lipids are transported along the subunit cavity, but the headgroups hop between the extracellular and intracellular entrances of the cavity, being only transiently present in its central part.

8. Does the ion conductive intermediate conformation also permeate phospholipids? If phospholipids are excluded from the pore in this state, it means the scramblases can form a protein-only ion conductive pores, which is contradictory to the proteolipidic pore model proposed in the study.

We apologize for this misunderstanding. Indeed, as we now show in Supplementary Fig. 6a, lipid headgroups are excluded from the central part of the nhTMEM16 cavity in the intermediate state, as we now discuss on lines 161–164. Similarly, the lipid headgroups were largely excluded from the pore of the L302A mutant (Supplementary Fig. 6b), in agreement with its impaired scrambling observed in experiments (Khelashvili et al, Nat. Commun., 2019). Nevertheless, both the mutant and the wild-type protein in the intermediate state reproducibly demonstrated much lower ion-conduction competence compared to that of the fully open conformation (Fig. 1e and Supplementary Fig. 2b). Based on these results we established the proteolipidic pore as the main (but not the only) ion-conductive state of TMEM16 lipid scramblases.

9. How membrane's electrical field affects the orientation of the permeation pathway's lining residues and lipid headgroups is intriguing. However, it would be more informative if the authors elucidate whether the lipid headgroups (PC, PS and PG) prefer upward movement in concurrence with their upward orientation in positive voltage (and vice versa in the negative voltage). The authors listed almost all charged residues in Fig. 7B, but missed out several conserved charged residues in Fig. 7C (D503, E506 for nhTMME16 and D497 and E500 for TMEM16K). Why? Are they im-

portant for ion conduction? Which directions do their sidechains point to?

In our simulations, we observed six complete lipid-scrambling events: three at positive and three at negative voltages. All the events were observed in the simulations of the fully open conformation of nhTMEM16 in a POPC membrane. All the scrambled lipids passed through the central part of the cavity, where they had their headgroups oriented according to the direction of the applied electric field. At negative voltages, all the lipids were transported from the inner to the outer leaflet of the membrane. At positive voltages, two lipids were transported in inward and one lipid in outward direction. Although it can be taken as a hint that headgroup orientation could be related to the transport direction, in the recent detailed study on the nhTMEM16 scrambling mechanism (Lee et al, Nat. Commun., 2018) no such connection was established. Thus, since the aim of our simulations was to study ion conduction and not lipid scrambling, our data do not allow for a strong conclusion on this matter, and we prefer to keep the focus of our manuscript on mechanisms of ion conduction rather than on lipid scrambling. The details of the observed scrambling events are now summarized in Supplementary Table 7 and discussed on lines 170–180.

The conserved acidic residues that were omitted in Fig. 7c form the Ca^{2+} -binding sites in nhTMEM16 and TMEM16K. None of these residues directly interact with the permeating ions in nhTMEM16 (Supplementary Fig. 10a,b) or TMEM16K (Supplementary Fig. 18c), suggesting no direct effect on the ion permeation process; we have now specified this on lines 373–376.

10. The authors emphasized that lipid and voltage can dictate the ion permeability and selectivity, what about the role of voltage in determining lipid selectivity, permeability and direction of movement?

As discussed in the response to the previous comment, our study was primarily designed to investigate ion permeation and its modulation by lipids, and our data do not permit definite conclusions about voltage effects on lipid transport in TMEM16 lipid scramblases. Although we assume that voltage would affect the transport of lipids depending on the charge of their headgroup, quantification of lipid selectivity would require a completely different simulation design outside the scope of this study; however, these are important questions to be addressed in future studies. Nevertheless, we do assume a stimulating effect of voltage at least on the scrambling of neutral lipids, as we discuss on lines 159–161.

11. The description about E313/E318/R432 interacting with lipid headgroups require a revision. Lee et al., Nat Comm 2018, which examined and described the roles of this triad in coordinating phospholipids, utilized POPE and POPG instead of POPC. Difference in lipid headgroups may affect the their interactions with the residues lining the permeation pathway.

Both POPE:POPG and POPC bilayers were used in the paper by Lee et al.: the important role of the E313/E318/R432 triad in lipid scrambling was first shown for nhTMEM16 in a POPE:POPG membrane and then confirmed via tICA analysis of multireplica simulations of nhTMEM16 in a POPC membrane.

12. This study underscores the importance of proteolipidic pore and lipid lining in determining ion permeability and selectivity. However, how would the author explain the ion conductive nhTMEM16 mutations, such as L302A (Khelashvili et al., Nat Comm 2019), that do not scramble lipids? How ion permeability and selectivity are defined in those non-scrambling cases? The non-scrambling mutations demonstrate that the widely open and proteolipidic pore might not be the only requirement for ion permeation in the scramblases as described by the authors.

We have now conducted simulations of the L302A mutant, which demonstrated lower ion-conduction competence (ion conductance and conduction probability; Supplementary Fig. 2) than wild type nhTMEM16 in the fully open conformation. The onset of ion conduction in the L302A

mutant was associated with an increase in pore width (Supplementary Fig. 4b), but the pore hydration remained similar in the conductive and nonconductive protomers (Supplementary Fig. 4c). In full agreement with the impaired scrambling demonstrated by the mutant (Khelashvili et al, Nat. Commun., 2019), lipid headgroups were largely excluded from the central part of its cavity (Supplementary Fig. 6b). Thus we conclude that the mechanism of ion conduction of the L302A mutant is very similar to that of the wild-type protein in the intermediate state, with ion permeability mainly defined by the pore width. The analysis of the mutant simulations is now shown in Supplementary Fig. 2, Supplementary Fig. 4, and Supplementary Fig. 6. These results are discussed on lines 110–114, 138–142, and 161–164.

Although the reduced ion conduction in the scrambling-incompetent state of nhTMEM16 hampers reliable analysis of its ion selectivity from unbiased submicrosecond MD simulations, our data hint that the pore is at least moderately Cl^- selective in the intermediate state (in total, 48 Cl^- and 29 Na^+ permeation events). We showed that the moderate chloride selectivity of the fully open state drastically changes upon addition of anionic lipids, which are located at the entrances to the pore and do not populate its center. Thus, since lipid headgroups have access to the entrances of the pore in the intermediate state, we believe that addition of anionic lipids to a membrane would lead to an increase in PNa/PCl , similar to the selectivity shift observed in the fully open state of nhTMEM16. We have added discussion of the ion selectivity of the non-scrambling pore on lines 480–482.

Although the nhTMEM16 pore in the scrambling-incompetent intermediate state is ion conductive, we would like to stress that it is the much higher ion-conduction potency of the proteolipidic pore that defines the fully open conformation as the main ion-conductive state of the nhTMEM16 lipid scramblase.

13. In general, the authors need to tune down their conclusions as neither their own findings nor previously published results fully support their conclusions.

We hope that the data we added during the revision will clarify the connection between our results and our conclusions. Below, we briefly describe the main conclusions together with the evidence that support them.

First, we showed that—as opposed to the fully open proteolipidic pore—nhTMEM16 in the intermediate state (i) demonstrates reduced ion conductance, (ii) requires notable conformational changes to initiate ion conduction, and (iii) has lower conduction probability. Based on these three lines of evidence, we concluded that the proteolipidic pore represents the main ion conductive state of Ca^{2+} -bound TMEM16 lipid scramblases.

Second, we demonstrated that (i) the lipid-headgroup orientation affects ion selectivity, (ii) the central lipid headgroup can control the permeability state of the pore, and (iii) the membrane lipid composition strongly affects ion selectivity of nhTMEM16. Based on these results we conclude that lipids have a pronounced effect on ion conduction properties of TMEM16 lipid scramblases. Finally, we demonstrated that human TMEM16K conducts ions through a cation-selective proteolipidic pore and exhibits weaker coupling between ion and lipid transport.

There are many minor issues in the manuscript. The reviewer only lists a couple here.

1. In Figure 5 and Figure 7, the authors used the numbers of protomers that display different PNa/PCl to compare their ion selectivity. While this is clear, it would be better to show the PNa/PCl values. Would the weighted PNa/PCl based on the number of protomers be calculated and presented?

We believe that the distribution of the protomers over the ion-selectivity classes is a rather clear and statistically sound way to demonstrate the effect of different conditions on the ion selectivity and its variability at the same time. Nevertheless, we have followed the reviewer’s suggestion and calculated weighted permeability ratios, which are now shown in Supplementary Table 8.

2. When the authors described their results in text, often times they only mentioned figure number without pointing to the exact panels. For instance, when referencing the supplementary figure 7, it would help the readers to refer to specific panels.

We thank the reviewer for this suggestion. We have now specified the panels of supplementary figures in the text more precisely.

Reviewer #3 (Remarks to the Author):

The manuscript describes the results of a very detailed investigation of ion permeability in TMEM16 scramblases (nhTMEM16, TMEM16K and TMEM16F) from the application of a method considered to represent “computational electrophysiology simulations”, termed CompEL. The findings support the proteolipidic-pore mechanism with a fully open groove as a main permeation route, against the alternating pore-cavity model. Detailed reports of the apparent role of lipid headgroups in shaping this ion pathway and controlling the permeability are likely to advance thinking in the field. Ion selectivity changes are shown to change with lipid composition or salt concentration and for TMEM16K, residue types in the groove and protein structure suggest that TMEM16K is a cation-selective ion channel.

The conclusions are sufficiently intriguing to engender interest, which should be sufficient to eliminate the for presenting them in a manner suggesting that the aim of the manuscript is to refute the connection between ion permeation and the intermediate state. In reality the prevalent view in the field based on specific evidence is that the intermediate state does not transport lipids but still can transport ions. The results presented in the manuscript are sufficiently interesting and do not require the establishment of a false dichotomy, so that a reconsideration of statements such as .. is recommended.

We thank the reviewer for finding our results interesting. Indeed, our results prove against the “alternating pore-cavity model”, since the scrambling-competent fully open state of the cavity is ion conductive. Moreover, in the fully open state ions permeate through a proteolipidic pore, which is characterized by higher overall ion-conduction competence compared to that of the intermediate state. In Supplementary Fig. 6a we now present data demonstrating that lipid headgroups are excluded from the central part of the pore in the intermediate state; these data are in agreement with the scrambling-incompetent ion-conductive pore of nhTMEM16 in the intermediate state.

We fully agree with the reviewer that the mentioned dichotomy—which was only unintentionally established—was unnecessary. Although the small ion conductance of the intermediate state was already appreciated in the original version of the manuscript, we have now rewritten the relevant sections of our manuscript (on lines 65–68, 420, and 433–441) to emphasize that both the intermediate and the fully open states can conduct ions, while the latter represents the main ion-conductive state.

Some concerns to be addressed:

A. The manuscript is based on the use of the CompEl molecular dynamics simulation method. This is not a widely known or used approach and thus requires much more definition and discussion of assumptions and considerations than is afforded in the manuscript. The very setup of the simulated system for which all the results are reported is provided only in the Supplementary Materials (Fig. 1a there). This alone raises two important issues that merit discussion and clarification in the manuscript A1. The title and text refers to computational electrophysiology. This may, or may not be an apt description of the method but a brief review of why and to what extent this approximation is considered to reproduce insights from electrophysiology experiments for this type of system (which may not be a gated channel of the kind discussed previously by this group) is in order, because the language describing the experiments is taken from actual electrophysiological measurements, which could be very misleading (e.g., line 82 “induced by the applied voltage”) when the simulation is carried out in an ion gradient established in a box).

We thank the reviewer for pointing this out. The ion/water exchange algorithm in Computational Electrophysiology (CompEL) simulations mimics the source of the electromotive force in electrophysiological experiments, where the voltage is applied by releasing/adsorbing Cl^- ions from the solution by Ag/AgCl electrodes. Thereby, CompEL simulations with sustained electrochemical potential gradients (by imposing a charge imbalance between the compartments separated by the membranes) permit the system to be studied under realistic transmembrane voltages (Kutzner et al, Biochim. Biophys. Acta, Biomembr., 2016). Since all our simulations are designed to mimic the conditions in electrophysiological experiments as close as possible, and since our analyses are aimed at obtaining functionally relevant electrophysiological parameters of ion conduction and selectivity, we believe that the usage of the electrophysiological language is justified in our case. The CompEL method is now discussed in more details on lines 82–89 and 611–614, and we have included more details into Supplementary Fig. 1, which illustrates the simulation setup.

A2. The setup description in Supplementary Fig. 1a shows that the EC and IC ends of the two proteins sampled are embedded in water compartments in which the cation is the same (both in the middle and at the extreme ends of the box). That, and the asymmetry of the membrane would be different in an electrophysiology experiment. While these may be acceptable approximations, they need to be presented as such. In particular, would this symmetry be expected to affect the influx/efflux results for the ions in Figure 4?

Similar to our simulations, patch-clamp experiments are often conducted under symmetric ionic conditions with the same cation being applied both intra- and extracellularly. In particular, Na^+ was symmetrically applied in a recent study demonstrating the ion-selectivity shift in TMEM16F in response to increased salt concentrations (Ye et al, eLife, 2019). Since we decided to focus on anion/cation selectivity, which is expected to be more sensitive to lipid effects compared with relative cation selectivities, we used only symmetric aqueous salt solutions to maximize convergence of our simulation results. We now clearly describe the ionic conditions and the inherent simplification on lines 590–591.

Indeed, given the effects of lipids on ion conduction, which we demonstrated in this study, we would expect the membrane asymmetry to affect the functional properties of TMEM16 lipid scramblases. However, under experimental conditions this asymmetry would be rapidly dissipated due to the presence of active scramblases in the membrane. Therefore, we believe that the symmetric membrane corresponds closer to the experimental conditions, when intracellular Ca^{2+} is applied to activate the lipid scramblases. This point is now discussed on lines 588–590. Finally, we expect that the lipid asymmetry would have no effect on the ion influx/efflux results, since it is the lipid headgroup at the central part of the pore (i.e. on the way between the membrane leaflets) that mainly controls the permeability state of the pore.

B. The manuscript refers to structures and mechanistic studies of nhTMEM16 in which intermediate and open groove states were described in detail. But the widening of the structure in the presented simulations is described by RMSD measures. Could this be shown in more details? For example, by calculating changes in pore radius and/or by following in the trajectories the distances between landmark residues Y439-T333, and between the E313, E318, and R432 triad residues that emerged as mechanistically essential. Indeed, it would be good to present structural details from the simulations in the manuscript regarding the following (i)- How the pair-wise distances between the residues in the E313/E318/R432 triad change in the simulations (ii)- What conformations these residues adopt when ions bind to the EC side and get transported (iii)-How the “widened” structure compares to the L302A structure of this TMEM16, and whether it conduct lipids (iv)- Whether lipid scrambling events were observed in the simulations and if so, what the relationship might be between the scrambling events and ion permeation events.

We have now added data on the pore width (measured as the distance between $\text{C}\alpha$ atoms of

Y439 and T333) to Supplementary Fig. 3c. Consistent with the increase in pore hydration, in the intermediate state the pore widens upon transition to an ion-conductive conformation. In this conformation, the pore is still narrower than that in the fully open state, precluding accommodation of the lipid headgroups in the central part of the pore, as now shown in Supplementary Fig. 6a. Summarizing, the widening of the pore is a prerequisite for ion conduction in the intermediate state, but the resulting width is not enough to permit lipid scrambling. This is now discussed on lines 134–136 and 435–437.

(i) We have measured the distances between the triad residues (R432–E313 and R432–E318) and found no correlation with the conduction state of nhTMEM16. We have added this analysis to Supplementary Fig. 8 and discuss it in the main text on lines 203–205.

(ii) While the R432–E313 and R432–E318 pairs sampled both interacting and noninteracting configurations, we did not find any effect of the occupancy of the extracellular ion-accumulation site on their pairwise distances in the simulations of nhTMEM16 in the fully open conformation (Fig. 1 for reviewers).

Figure 1 for reviewers: Distance between R432 and E313 or E318 measured separately for the frames, in which the extracellular site was empty (emp) or occupied (occ) by the indicated ion.

(iii) Since the structure of the intermediate state of the wild type (PDB ID: 6QMA) is structurally very similar to that of the L302A mutant (PDB ID: 6OY3), the widened ion-conductive pore of the intermediate state observed in simulations is also wider than that of the cryo-EM structure of the mutant. We have now indicated the width of the mutant in Supplementary Fig. 3c. Moreover, additional simulations of the L302A mutant demonstrated that initiation of ion conduction involves widening of the pore similar to the wild type (Supplementary Fig. 4b), as discussed on lines 141–142.

(iv) Since lipid scrambling is a much slower process compared with ion conduction (Supplementary Fig. 9) and since our simulations were designed to obtain optimal sampling of ion transport, only a moderate number of scrambling events was observed in our unguided MD simulations: We detected six complete scrambling events mediated by nhTMEM16 in the fully open state; these are now summarized in Supplementary Table 7 and discussed on lines 170–180. All the scrambled

lipids crossed the membrane along the subunit cavity and went through the central part of the pore.

Since we demonstrated that the lipid headgroups at the center of the pore can affect its permeability state, a certain level of interference may exist between lipid scrambling and ion permeation. However, the small number of observed lipid-scrambling events does not permit robust assessment of the effect of ion permeation on lipid scrambling. Nevertheless, our results indicate that there is likely no significant competition between lipid and ion transport, since the fully open cavity is more conductive than the one in the scrambling-incompetent intermediate state. Moreover, we showed that ion binding to the extracellular sites does not affect configuration of the E313/E318/R432 triad (see part ii of this comment); as the triad was shown to play an important role in the scrambling mechanism (Lee et al, Nat. Commun., 2018), we suggest that possible effects of permeating ions on lipid scrambling, if any, could only be explained by direct ion–lipid interactions but not by altered protein dynamics. The precise coupling between ion and lipid transport is an important question to be addressed in future studies.

C. A key point made in the manuscript concerns lipid head group orientations in negative/positive voltages. While it is clear that the lipid headgroups would change directions in the different potential fields it would be informative for a description of a scramblase mechanism to connect these results to the dynamics of lipid movement to clarify the interpretation. Incidentally, it appears that the error bars in Figure 2d are large and the data for zero and negative potentials basically overlaps. How can this be interpreted?

Indeed, the distributions of the angle between a headgroup and the membrane normal at zero and negative voltages overlap within the error bars in the intracellular and central parts of the pore, suggesting that at zero voltage a central lipid headgroup prefers downward orientation, as can be additionally seen from the Supplementary Fig. 7a.

Since the first part of this comment concerns the same matter as the comment #9 by reviewer 2, we refer here to the response to the latter. Briefly, our data hint at a correlation between headgroup orientation and direction of lipid scrambling, but rigorous establishment of this connection requires further investigation, which would be outside the scope of the current manuscript on ion conduction by the scramblases. This is now discussed on lines 170–180.

D. Was the analysis for Figure 2 done only for the fully open state (starting from 4WIS). It seems necessary to do it as well for the intermediate state (starting 6QMA), because conductive protomers were identified there as well (Figure 1) and information is needed regarding the lining of the pore with lipid headgroups in order to generalize about the requirement for a proteolipidic pore to observe ion permeation, as done in the manuscript. The same reasoning would suggest the need for analysis of the intermediate state to reach the conclusions regarding lipid realignment from Figure 3.

We have now added the analysis of the lipid-headgroup distribution along the pore in the intermediate state to Supplementary Fig. 6a. Indeed, lipid headgroups were excluded from the central part of the pore. Although we established the proteolipidic pore as the main (but not the only) ion-conductive state of the pore, we now prevent the wrong impression that the protein-only pore in the intermediate does not conduct ions at all. We have also analyzed the distribution of permeating ions through the pore in the intermediate state. These data are now shown in Supplementary Fig. 11a,b and discussed on lines 229–230 and 234–236.

E. The intent and logic of the following phrase in the Discussion seems unclear to this reader. Is the idea that fully activated TMEM16 are some sort of “super-channels”? “Given that in our simulations single-channel conductance 374 of nhTMEM16 in the intermediate state (10 pS) was comparable to that of TMEM16A 375 (3 - 8 pS), it is tempting to speculate that ion conduction in the intermediate state 376 resembles that of bona fide channels of the family, whereas a fully open subunit cavity is 377 necessary for complete activation of ion conduction in TMEM16 lipid

scramblases.”

We apologize for this confusion. At equilibrium Ca^{2+} -bound TMEM16 lipid scramblases are believed to adopt three different conformations: closed, intermediate and fully open. We demonstrated that these states are characterized by different levels of ion conduction, with the ion conductance of the intermediate state being close to the single-channel conductance of TMEM16A, which is a bona fide channel. Considering the common evolutionary origin of the channels and lipid scramblases of the TMEM16 family, our intent was to point at a possible connection between these two classes in terms of ion conduction; we now clarified this on lines 453–455.

F. An idea presented in the Discussion as a hypothesis appears to be a guess or speculation not really based on anything specific:

“We hypothesize

427 that the deranged Ca^{2+} signaling leading to spinocerebellar ataxia in patients with
428 mutations in TMEM16K41 could, at least partly, result from impaired function of TMEM16K
429 as ER-localized cation-selective ion channel..

This is followed by a further speculation about the consequences. Unless labeled as a speculation, some clarification for the basis for this hypothesis would seem to be necessary here.

We thank the reviewer for pointing this out. We agree that this part of the discussion is speculative. Our intent was to highlight this possible, physiologically important, implication of our results and to inform further studies on TMEM16K as an ion channel. We have now clearly marked this part of the discussion as speculation on lines 512–513.

Reviewer #1 (Remarks to the Author):

All of my questions and concerns are addressed properly.

Reviewer #2 (Remarks to the Author):

This is a much improved version. Appreciate the authors' efforts. There are a few remaining concerns.

1. In most figures, there are no statistical tests to support the quantitative interpretation of the simulations. While data point containing bar graphs are shown in numerous figures, only Figure 4, Figure S12, and S19 contain statistical analyses. What warranted the use of statistical tests in these figures and not the others? For example, in Figure 1d-f, the authors discussed the conductance values and the number of water molecules between the intermediate state and the fully open state. However, there were no statistical analyses done to support the differences between the intermediate and full open states. While the probability of ion permeation is clearly different between the intermediate and fully open states, lack of rigorous statistical analyses undermines the authors' claim that the fully open conformation represents the main ion-conductive state of TMEM16 scramblases.
2. In figure 2a,b, the authors suggested that the choline headgroup ne (which belongs to the middle lipid with the pc headgroup) is coordinated by E313 and T333. However, while it appears to contact E313, it looks like it is still far away from T333. Please report the distances between the ne choline headgroup and T333 in both negative and positive voltages?
3. The observation that lipid headgroups can form steric hindrance to prevent ion permeation (in Figure 4) is interesting. However, one major concern is the lack of effects on ion influx between the open and closed state (Figure 4). If the claim that the open state is the key conductive state, these results strongly argue against this hypothesis. Please explain the discrepancy.
4. The analysis of the extracellular entrance of TMEM16K structure is interesting, but it should be taken with cautious when trying to generalize this with other scramblases, especially TMEM16F. It's because TMEM16K lacks that 'extracellular' (facing the lumen of the ER) ectodomain that is seen in both TMEM16A and TMEM16F. Please make corresponding changes in the discussion

Reviewer #3 (Remarks to the Author):

The concerns in my review were adequately addressed by the answers and the corresponding revisions of the manuscript, including the results of a highly relevant new simulation of the L302A mutant of nhTMEM16. I have no doubt that the work and the results add much needed insight and comparative information about the mechanisms of ion conductance by these proteins under various conditions.

In my opinion, a large and comprehensive study there will always arise details that differ from conclusions proposed by results obtained by other, based on what may or may not be the same types of measurements under conditions that may or may not be relevant to the model established and quantified computationally.

That such discrepancies appear to some readers is not a flaw, but a strength of a serious and clearly described and documented study as I consider the present one to be.

Harel Weinstein

Response to the reviewers' comments

Reviewer #1 (Remarks to the Author):

All of my questions and concerns are addressed properly.

We thank the reviewer once again for thoroughly reading our manuscript.

Reviewer #2 (Remarks to the Author):

This is a much improved version. Appreciate the authors' efforts. There are a few remaining concerns.

1. In most figures, there are no statistical tests to support the quantitative interpretation of the simulations. While data point containing bar graphs are shown in numerous figures, only Figure 4, Figure S12, and S19 contain statistical analyses. What warranted the use of statistical tests in these figures and not the others? For example, in Figure 1d-f, the authors discussed the conductance values and the number of water molecules between the intermediate state and the fully open state. However, there were no statistical analyses done to support the differences between the intermediate and full open states. While the probability of ion permeation is clearly different between the intermediate and fully open states, lack of rigorous statistical analyses undermines the authors' claim that the fully open conformation represents the main ion-conductive state of TMEM16 scramblases.

We thank the reviewer for pointing this out. We have now statistically tested all differences between various protein states which are discussed in the text. The results of the statistical analyses are added to Fig. 1d-f, Supplementary Fig. 3a-d, Supplementary Fig. 4b-c, and Supplementary Fig. 8a-b. They fully support our conclusions on differences in ion conductance, structure, and hydration between different states of nhTMEM16. In particular, the ion conductance is significantly higher and ion-permeation onset is significantly lower in the fully open compared with the intermediate conformation (Fig. 1d,e, Supplementary Fig. 3d). We have now also demonstrated that a statistically significant conformational change accompanies transition of protomers in the intermediate conformation to the ion-conducting state, while no such change is observed in nonconductive protomers (Supplementary Fig. 3a-c). Furthermore, a significant increase in pore hydration is shown for wild-type nhTMEM16 (Fig. 1f) but not for the L302 mutant (Supplementary Fig. 4c). Finally, our analysis did not detect statistically significant conformational differences of the R432/E313/E318 triad in conductive and nonconductive protomers, consistent with our conclusions (Supplementary Fig. 8a-b).

2. In figure 2a,b, the authors suggested that the choline headgroup ne (which belongs to the middle lipid with the pc headgroup) is coordinated by E313 and T333. However, while it appears to contact E313, it looks like it is still far away from T333. Please report the distances between the ne choline headgroup and T333 in both negative and positive voltages?

Multiple replicas of independent protomer trajectories were used for the analysis of contacts between the headgroup moieties and protein residues shown in Supplementary Fig. 7b and illustrated in Fig. 2a,b. Although Fig. 2a,b illustrate the typical arrangement of lipid headgroups within the cavity, a particular positioning of the protein residues and lipid headgroups shown in the figure can slightly deviate from its average, since a snapshot can only present an instantaneous state of the proteolipidic pore. To demonstrate the close proximity of the *ne* choline group to T333 and E313, we calculated distances from the nitrogen atom of the headgroup to oxygen atoms of the residues at positive and negative voltages; the distance histograms are shown in Figure 1 for reviewers. They clearly demonstrate that both T333 and E313 form contacts with the *ne* choline group in comparable number of simulation frames (overall mean probability of about 50 %, see Supplementary Fig. 7b).

Figure 1 for reviewers: Distribution of distance between nitrogen atom of the *ne* choline and oxygen atoms of T333 and E313 in all simulated protomers at positive and negative voltages. Dashed line indicates the distance threshold used in the contact definition.

3. *The observation that lipid headgroups can form steric hindrance to prevent ion permeation (in Figure 4) is interesting. However, one major concern is the lack of effects on ion influx between the open and closed state (Figure 4). If the claim that the open state is the key conductive state, these results strongly argue against this hypothesis. Please explain the discrepancy.*

Although lipid headgroups demonstrated much more pronounced blocking effect on ion efflux than on ion influx, we note that at least Cl^- influx was statistically significantly affected by the extracellular phosphate groups at the *pe* site (Fig. 4a). However, both these open and closed (blocked) microstates of the pore in this analysis relate to the fully open conformation, which represents the main ion-conductive state. Therefore, neither the lack nor presence of the blocking effect would contradict our conclusions. We apologize for this confusion and have added a clarification to the figure legend and on line 260 of the main text.

4. *The analysis of the extracellular entrance of TMEM16K structure is interesting, but it should be taken with cautious when trying to generalize this with other scramblases, especially TMEM16F. It's because TMEM16K lacks that 'extracellular' (facing the lumen of the ER) ectodomain that is seen in both TMEM16A and TMEM16F. Please make corresponding changes in the discussion*

We agree that generalization of the TMEM16K properties to the rather distant isoform TMEM16A and TMEM16F must be done with caution. Although we neither did nor implied such generalization in the manuscript, we have added a note on the ectodomain to the text on lines 376–378.

Reviewer #3 (Remarks to the Author):

The concerns in my review were adequately addressed by the answers and the corresponding revisions of the manuscript, including the results of a highly relevant new simulation of the L302A mutant of nhTMEM16. I have no doubt that the work and the results add much needed insight and comparative information about the mechanisms of ion conductance by these proteins under various conditions. In my opinion, a large and comprehensive study there will always arise details that differ from conclusions proposed by results obtained by other, based on what may or may not be the same types of measurements under conditions that may or may not be relevant to the model established and quantified computationally. That such discrepancies appear to some readers is not a flaw, but a strength of a serious and clearly described and documented study as I consider the present one to be.

Harel Weinstein

We thank Dr. Weinstein for his positive evaluation of our work.